# ToolTree: Efficient LLM Agent Tool Planning via Dual-Feedback Monte Carlo Tree Search and Bidirectional Pruning

**Shuo Yang[1], Soyeon Caren Han[1], Yihao Ding[2], Shuhe Wang[1], Eduard Hovy[1]**
[1]School of Computing and Information Systems, The University of Melbourne
[2]School of Physics, Mathematics and Computing, The University of Western Australia
`shuo.yang.3@unimelb.edu.au`

## Abstract

Large Language Model (LLM) agents are increasingly applied to complex, multi-step tasks that require interaction with diverse external tools across various domains. However, current LLM agent tool planning methods typically rely on greedy, reactive tool selection strategies that lack foresight and fail to account for inter-tool dependencies. In this paper, we present ToolTree, a novel Monte Carlo tree search-inspired planning paradigm for tool planning. ToolTree explores possible tool usage trajectories using a dual-stage LLM evaluation and bidirectional pruning mechanism that enables the agent to make informed, adaptive decisions over extended tool-use sequences while pruning less promising branches before and after the tool execution. Empirical evaluations across both open-set and closed-set tool planning tasks on 4 benchmarks demonstrate that ToolTree consistently improves performance while keeping the highest efficiency, achieving an average gain of around 10% compared to the state-of-the-art planning paradigm.[1]

## 1 Introduction

Recent advancements in Large Language Models (LLMs) (Brown et al., 2020; Ouyang et al., 2022; Touvron et al., 2023) have propelled the emergence of language agents capable of tackling complex multi-step tasks across various domains, including software engineering (Yang et al., 2024), web browsing (Zhou et al., 2023), scientific discovery (Bran et al., 2023) and multimodal understanding (Wu et al., 2023). A critical aspect of enabling these agents to solve sophisticated problems lies in their ability to plan and coordinate external tools (Qu et al., 2025). Effective tool planning leverages the prior knowledge of LLMs by decomposing complex tasks, reasoning about which tools are appropriate, and generating structured plans that assign intermediate steps to these tools. In doing so, LLMs can integrate external functionalities into their reasoning process, thereby enhancing their effectiveness in completing complex tasks (Schick et al., 2023; Li et al., 2024; Lu et al., 2025).

To enhance the tool planning capabilities of LLMs, existing research has primarily followed two directions. The first is greedy-based tool planning, where the model independently selects and executes the tool that appears most suitable at each step, without engaging in long-term rewards. (Wei et al., 2022; Shen et al., 2023; Yao et al., 2023b; Lu et al., 2025; Liu et al., 2025). As a result, these approaches often suffer from brittle performance, particularly when early suboptimal choices propagate errors that compound irreversibly and compromise later steps. Besides, these methods also tend to waste computation by following only a single trajectory with no exploration of alternatives. On the other hand, search-based methods attempt to address this limitation by expanding multiple candidate branches, but they introduce new challenges when tools are involved (Yao et al., 2023a; Zhuang et al., 2024; Zhou et al., 2024). The branching factor grows exponentially with tool types, arguments, and evolving states, leading to high costs and unpredictable latency. Moreover, many variants evaluate hypothetical thoughts rather than executed actions, so ranking is decoupled from actual tool use utility, and improvements realized several steps later are rarely credited back to

---

[1]Code: `https://github.com/SYang2000/ICLR_2026_ToolTree/tree/main`

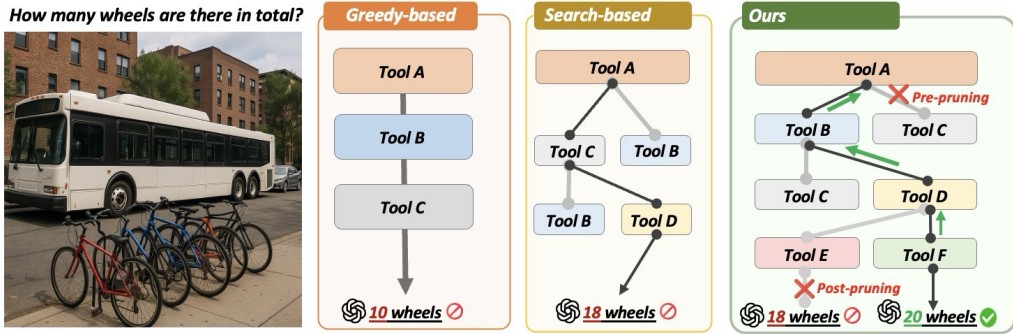

Figure 1: Comparison of ToolTree with greedy search and search-based tool planning. Our ToolTree chooses the optimal tool trajectory and answers correctly with 20.

earlier decisions. Together, these drawbacks highlight the need for a planning approach that is both forward-looking and outcome-grounded, while remaining compute-efficient under fixed budgets.

Our ToolTree tackles the above two issues at the same time. ToolTree frames tool planning as a search problem guided jointly by a fast pre-execution prior and a grounded post-execution utility, enabling agents to allocate computation adaptively and recover from early missteps without task-specific retraining as illustrated in Figure 1. Our design integrates pre-execution scoring into the selection policy to predict the utility of a tool before it is invoked, while a post-execution score assesses its actual contribution based on observed outcomes as rollout rewards, and applies complementary pre- and post-pruning to eliminate unpromising branches. This feedback loop enables the agent to refine its strategy iteratively, incorporating foresight and hindsight into tool selection. To evaluate the effectiveness of ToolTree in enhancing LLM agent tool planning abilities, we compare ToolTree with greedy and search-based planning methods on four tool use benchmarks spanning both closed-set and open-set tool scenarios, with around 10 percent improvement over baseline, achieving SoTA performance with a 66.95 F1 score on GTA and a 69.04 pass rate on ToolBench.

Overall, our contributions can be summarized as follows:

- We present ToolTree, a novel Monte Carlo tree search-inspired planning paradigm that frames LLM agent tool use as search guided by pre-execution priors and post-execution rewards, enabling robust multi-step reasoning without retraining.
- ToolTree effectively integrates a dual-evaluation guided tree traversal method with bidirectional pruning, which integrates pre- and post-scoring into search and eliminates weak branches, improving accuracy per unit compute under fixed budgets.
- We evaluate ToolTree on four benchmarks of both closed-set and open-set tool planning, demonstrating its superior effectiveness and efficiency. The improvements scale consistently with the number of tool sets, model size and computing resources.

## 2 PRELIMINARIES

In this section, we introduce the preliminaries of the tool planning task for language agents, including (1) the formal problem definition for tool planning; (2) tree-search enhanced tool planning; and (3) the fundamentals of Monte Carlo Tree Search (MCTS).

**Problem Definition: Tool Planning.** Tool planning is the task that decides not only which tools a language model should use, but also when and in what order to use them, in order to accomplish a task both efficiently and accurately. Unlike simple tool selection, which focuses on identifying the most appropriate tool at a single step, tool planning requires reasoning over entire sequences of tools, with the objective of discovering an optimal sequence that maximizes task success.

Formally, let (1) $\mathcal{T}_{\text{lib}} = \{t_1, t_2, \ldots, t_m\}$ denote a set of available tools. Each tool $t \in \mathcal{T}_{\text{lib}}$ is represented by a structured tool card $C_t$ with explanatory metadata using JSON format to provide

standardized information for further utilization, which can be found in Appendix B.5; (2) $S$ be the state space, where each state $s \in S$ encodes the current dialogue context and any accumulated intermediate results; (3) $A$ denote the action space, where each action corresponds to invoking a tool $t_i \in \mathcal{T}_{\text{lib}}$ with an input; and (4) $R : S \rightarrow \mathbb{R}$ be a reward function that measures how correct, informative or efficient the current tool sequence is. Then, the tool planning task is to learn or search for a policy $\pi : S \rightarrow A$ that generates a sequence of actions $s^* = \{a_1, a_2, \ldots, a_n\}$, where $a_i \in A$ to maximize the expected reward: $\pi = \arg \max \mathbb{E}[R(s^*)|\pi, \mathcal{T}_{\text{lib}}]$

**Tree Search-enhanced planning.** Tree-search enhanced tool planning reframes the above tool planning task as a search problem: The agent explicitly constructs and evaluates candidate sequences of tool invocations (sequences) and uses a specific search policy to choose actions that are promising in expectation. Specifically, a search tree is constructed with nodes corresponding to states $s \in S$ and edges corresponding to actions $a \in A$. Each root-to-node path corresponds to a partial plan $s = \{a_1, \ldots, a_k\}$, where $k$ denotes the number of searched child nodes. The tree search procedure estimates terminal rewards $R(s^*)$ for candidate plans and returns the highest-value plan.

**Monte Carlo Tree Search (MCTS).** Monte Carlo Tree Search (MCTS) is a heuristic search algorithm for decision-making in large and complex search spaces, most notably applied in game playing (e.g., Go (Silver et al., 2016) and Chess (Helfenstein et al., 2024)) and planning problems (Feng et al., 2023). Basically, the MCTS process can be decomposed into four iterative steps: (1) selection, starting from the root, the algorithm recursively selects child nodes according to a tree policy, such as UCT (Kocsis & Szepesvári, 2006) and PUCT (Silver et al., 2017) that balances exploration and exploitation; (2) expansion, if the selected node is not terminal, one or more child nodes are added to the tree, representing possible future actions; (3) simulation, from the expanded node, a policy-guided simulation is performed to approximate the outcome of completing the plan from that state; (4) back propagation, the result of the simulation is propagated back up the tree to update the computed rewards of the traversed nodes. By repeating this procedure many times, MCTS refines its estimates of action values and converges toward high-quality plans.

## 3 PROPOSED METHOD: TOOLTREE

In this section, we demonstrate how ToolTree performs tool planning by casting multi-tool use as a Monte Carlo Tree Search (MCTS) inspired planning paradigm over executable trajectories in Figure 2. We first outline the overall process in Section 3.1. We then demonstrate the unique design of dual evaluation and pruning in Section 3.2.

### 3.1 OVERVIEW

We view tool planning as a sequential decision process where each state encodes the evolving dialog context and intermediate results, and each action corresponds to invoking a candidate tool from the library $\mathcal{T}_{lib} = \{t_1, \ldots, t_m\}$. The objective is to discover a trajectory that maximizes task utility within a fixed rollout budget $R_{\max}$.

Unlike prior approaches that rely on a separate planner, *ToolTree* integrates tool selection, execution, evaluation and pruning directly into the MCTS loop. At every step, the search is guided by two complementary signals: a lightweight *pre-evaluation* that anticipates the usefulness of an action before execution, and a *post-evaluation* that scores the grounded output afterward. This dual feedback supports both exploration and pruning, enabling deliberate, training-free planning that generalizes across diverse tool libraries. The search terminates when the budget is met or improvements plateau, and the highest-valued trajectory is returned to generate the final answer. This look-ahead/look-back loop allows the agent to recover from early errors, avoid dead-end tool combinations, and allocate its limited call budget to the most promising trajectories. The overall process is depicted in Figure 2 with Selection, Pre-evaluation, Expansion, Execution, Post-Evaluation, and Backward Propagation.

**Selection.** Given the current search state $s$, the search descends the tree by repeatedly selecting the child action that maximizes a *prior-augmented* UCT score:

$$\text{UCT}(s, a) = Q(s, a) \; + \; \lambda \, r_{\text{pre}}(s, a) \sqrt{\frac{\ln N(s)}{N(s, a)}}. \tag{1}$$

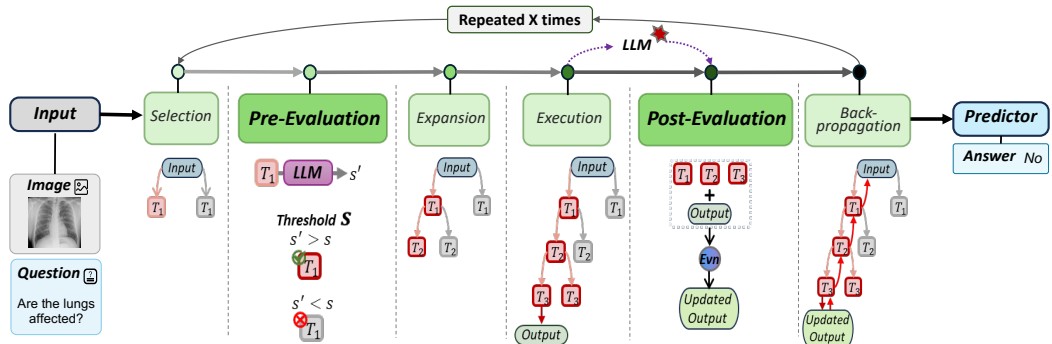

Figure 2: **Architecture overview of ToolTree**. An input query is processed sequentially via iterative *dual evaluation-guided Monte Carlo Tree Search*, including selection, pre-evaluation, expansion, execution, post-evaluation and backward-propagation. The *Answer Predictor* then incorporates the tool trajectories with the highest reward found by the MCTS to produce the final prediction.

where $Q(s,a)$ drives exploitation as it accumulates the *post evaluation* rewards obtained so far. $N(s)$ and $N(s,a)$ are visit counts and $r_{\mathrm{pre}}(s,a) \in [0,1]$ is a fast, predictive signal available *before* executing action $a$. Only admissible actions $a \in \mathcal{A}(s)$ where tools with input schema compatible with the current context are considered; ties are broken by larger $N(s)$ followed by a small random jitter to preserve exploration diversity. The use of $r_{\mathrm{pre}}(s,a)$ biases early rollouts toward promising branches while retaining the exploitation pressure from $Q(s,a)$.

**Expansion.** Upon reaching a leaf state $s_t = \langle C_t, A_{1:t} \rangle$, we enumerate the remaining admissible actions $\mathcal{A}_{\mathrm{rem}}(s_t) = \mathcal{A}(s_t) \setminus \{a_1, \ldots, a_t\}$. For each candidate $a \in \mathcal{A}_{\mathrm{rem}}(s_t)$, we obtain its predictive score $r_{\mathrm{pre}}(s_t, a)$ and instantiate a new child node $(s_t, a)$ only if $r_{\mathrm{pre}}(s_t, a) \geq \tau_{\mathrm{pre}}$ (pre-pruning, consistent with the prior term in Eq. 1), and the tool's I/O schemas are type-compatible with $C_t$. When tools accept structured arguments, we generate a minimal, schema-valid argument draft and cache it with the node to avoid regenerating at selection time.

**Execution.** For a selected child $(s_t, a)$, we invoke the corresponding tool/API with its arguments, yielding an output $o_{t+1}$. The context is updated to $C_{t+1}$ by appending $(a, o_{t+1})$ in a structured form. To reduce waste, we employ deterministic caching keyed by $(a, \mathrm{args})$: if an identical call has already been made within the current rollout, its $o_{t+1}$ is reused. Persistent failures attach an error token to $o_{t+1}$ so downstream compatibility checks and scoring can handle the outcome explicitly.

**Backward Propagation.** After execution, the resulting post-execution score $r_{\mathrm{post}}(s_t, a) \in [0,1]$ is propagated from the new child back to the root. For every edge $(s, a)$ on this path, we update the counts and value estimate $N(s,a) \leftarrow N(s,a) + 1, \qquad Q(s,a) \leftarrow Q(s,a) + \frac{r_{\mathrm{post}}(s_t,a) - Q(s,a)}{N(s,a)}$. This running average refines the exploitation term in Eq. 1, allowing subsequent selections to reflect observed utility. We also maintain $N(s) \leftarrow \sum_{a'} N(s, a')$ for use in the exploration bonus.

## 3.2 DUAL EVALUATION AND PRUNING

Classical MCTS balances exploration and exploitation but is agnostic to (i) the *plausibility* of a tool call before execution and (ii) the *grounded utility* of its realized output afterwards. *ToolTree* injects two lightweight, training-free signals into the loop: a *pre-evaluation* $r_{\mathrm{pre}}(s,a) \in [0,1]$ that forecasts usefulness prior to execution, and a *post-evaluation* $r_{\mathrm{post}}(s,a) \in [0,1]$ that scores the produced output. These signals serve complementary roles—foresight and hindsight—and enable *bidirectional pruning* that keeps the tree compact without sacrificing solution quality.

**Pre-Evaluation.** For a newly encountered pair $(s, a)$, we query a LLM judge to score $r_{\mathrm{pre}}(s,a)$ based on the current context $C$, the tool card (I/O schema, domain tags, examples), and a schema-valid argument draft. This score enters selection via the prior-augmented exploration bonus in Eq. 1 and also gates expansion:

$$\mathcal{A}^+(s_t) = \{ a \in \mathcal{A}(s_t) : r_{\mathrm{pre}}(s_t, a) \geq \tau_{\mathrm{pre}} \}, \qquad \mathcal{A}_{\mathrm{keep}}(s_t) = \text{top-}K(\mathcal{A}^+(s_t); r_{\mathrm{pre}}).$$

Only actions in $\mathcal{A}_{\text{keep}}(s_t)$ are expanded. Intuitively, $r_{\text{pre}}$ removes obviously incompatible or low-yield branches *before* any tool call, reducing the branching factor while still allowing exploration through the UCT term. Depth-aware annealing of $\lambda$ (or of $\tau_{\text{pre}}$) can gradually temper the influence of the prior as empirical evidence accumulates.

**Post-Evaluation** After executing $(s_t, a)$ and obtaining $o_{t+1}$, we score grounded utility with the same LLM judge:

$$r_{\text{post}}(s_t, a) \;=\; J\big(C_t, \, a, \, o_{t+1}\big) \in [0, 1],$$

where $J$ evaluates task-consistency (e.g., correctness proxies, relevance, constraint satisfaction) and robustness cues. This score drives exploitation by updating the running mean $Q(s, a)$ in backward propagation and directly supports *post-pruning*: edges with $r_{\text{post}}(s_t, a) < \tau_{\text{post}}$ are marked non-expandable to prevent further budget on unproductive continuations. Because $r_{\text{post}}$ is computed on *executed* actions, it yields faithful credit assignment compared to ranking hypothetical thoughts.

**Bidirectional Pruning.** Combining both signals yields a two-sided budget control: Pre-pruning to discard $(s, a)$ if $r_{\text{pre}}(s, a) < \tau_{\text{pre}}$ (or if it falls outside the top-$K$),thereby curbing expansion of low-promise children. Post-pruning while after execution, mark $(s_t, a)$ non-expandable if $r_{\text{post}}(s_t, a) < \tau_{\text{post}}$, trimming branches disproven by evidence. We also cache $(a, \text{args}) \mapsto o$ to avoid duplicate calls within a rollout; failures attach a typed error token so pruning decisions remain explicit rather than implicit timeouts. Together, these rules concentrate rollouts on branches that are both *likely* (per $r_{\text{pre}}$) and *useful* (per $r_{\text{post}}$), improving accuracy-per-second under fixed $R_{\max}$.

## 4 EXPERIMENTS

### 4.1 EXPERIMENT SETUP

We evaluate *ToolTree* across two complementary regimes that stress different facets of LLM agent tool use: *closed-set tool planning* with GTA (Wang et al., 2024) and m&m (Ma et al., 2024), where a small, fixed tool set with typed I/O must be composed into short multi-hop chains, and *open-set tool planning* with ToolBench (Qin et al., 2023) and RestBench (Song et al., 2023), where the action space spans dozens of APIs/endpoints and API retrieval is part of the problem. These two tasks demonstrate the effectiveness and efficiency of ToolTree.

**Datasets.** We use four datasets covering two tasks to test our method. For (i) *closed-set tool planning* we adopt **GTA** (Wang et al., 2024) and **m&m** (Ma et al., 2024), each of them provides a fixed tool set of size 14/33 with typed I/O and short multi-hop chains. We follow the original setup by evaluating this task in both step-by-step mode and end-to-end modes. For (ii) *open-set tool planning* we use **ToolBench** (Qin et al., 2023) and **RestBench** (Ma et al., 2024), which pair 16,464 and 143 real APIs, respectively, with multi-tool retrieval-then-planning scenarios under a judge-based protocol. We follow the initial setup using pass rate and win rate as the evaluation metrics. More details can be found in Appendix B.1 and B.2.

**Baselines.** For (i) *closed-set tool planning* on GTA and m&m we compare our method against: 1) Zero-shot, 2) ReAct (Yao et al., 2023b), 3) Chain-of-Thought (Wei et al., 2022), 4) Best–First search (Koh et al., 2024), 5) Tree-of-Thought (Yao et al., 2023a), 6) A* Search developed in the ToolChain* paper (Zhuang et al., 2024), and 7) Monte Carlo tree search (Zhou et al., 2024). These baselines span the spectrum from no planning through greedy, reactive planning to search-based planning, providing a comprehensive contrast on small, typed tool suites. For (ii) *open-set tool planning* on ToolBench and RestBench, we use: 1) Zero-shot, 2) Chain-of-Thought, 3) ReAct, 4) DFSDT Qin et al. (2023), and 5) Monte Carlo tree search, emphasizing planning-centric controllers standard for large API spaces, while retaining simple baselines to isolate planning gains. To ensure a fair comparison, all planners share the same tool schemas and descriptions, the same type pre-gating pipeline and the same caching policy for tool outputs and LLM calls. We also enforce identical compute and rollout budgets. These shared engineering settings are applied uniformly across methods to isolate the effect of the planning strategy itself. More details in Appendix B.3.

| Model | Planner | GTA | | | | | m&m | | | | |
|-------|---------|-----|-----|-----|-----|-----|-----|-----|-----|-----|-----|
| | | Step-by-step | | End-to-End | | AVG | Step-by-step | | Multi-step | | AVG |
| | | Tool | Arg | Plan | Exec | | Tool | Arg | Plan | Exec | |
| GPT-4o-mini | Zero-Shot | 58.73 | 28.44 | 60.18 | 33.85 | 45.30 | 72.48 | 67.44 | 77.48 | 67.59 | 71.25 |
| | ReAct | 60.13 | 29.43 | 68.26 | 34.80 | 48.16 | 73.55 | 65.10 | 82.16 | 69.42 | 72.56 |
| | CoT | 56.10 | 25.58 | 66.47 | 35.63 | 45.95 | 70.13 | 65.27 | 76.12 | 66.96 | 69.62 |
| | Best–First | 58.42 | 30.13 | 69.96 | 34.46 | 47.99 | 74.42 | 66.83 | 83.58 | 68.37 | 73.80 |
| | ToT | 62.41 | 33.12 | 72.94 | 37.42 | 51.47 | 75.58 | 70.84 | 82.58 | 71.37 | 75.59 |
| | A* | 64.47 | 35.26 | 73.86 | 38.16 | 52.94 | 75.16 | **71.85** | 84.74 | 72.59 | 76.59 |
| | LATS | 65.88 | 37.26 | 74.28 | 38.24 | 53.91 | 76.84 | 70.16 | 83.38 | 72.94 | 75.83 |
| | **ToolTree (ours)** | **67.83** | **39.64** | **76.44** | **39.65** | **55.89** | **77.25** | 71.26 | **85.52** | **73.58** | **76.90** |
| GPT-4o | Zero-shot | 70.16 | 38.52 | 77.14 | 45.28 | 57.78 | 78.52 | 80.17 | 85.17 | 78.47 | 80.58 |
| | ReAct | 71.42 | 40.58 | 75.52 | 46.33 | 58.46 | 83.58 | 81.24 | 84.42 | 76.58 | 81.46 |
| | CoT | 66.52 | 42.17 | 73.22 | 42.86 | 56.69 | 85.58 | 77.84 | 78.16 | 71.43 | 78.75 |
| | Best–First | 72.13 | 44.26 | 77.64 | 47.83 | 60.46 | 84.47 | 82.17 | 85.84 | 78.11 | 82.65 |
| | ToT | 72.53 | 43.68 | 78.84 | 46.53 | 60.40 | 86.28 | 83.74 | 85.26 | 80.35 | 83.91 |
| | A* | 74.29 | 47.58 | 79.96 | 46.26 | 62.52 | 87.17 | 83.44 | 86.87 | 81.49 | 84.74 |
| | LATS | 77.84 | 49.90 | 82.57 | 48.80 | 64.78 | 88.89 | 84.77 | 88.38 | 83.77 | 86.45 |
| | **ToolTree (ours)** | **79.26** | **50.84** | **85.53** | **52.17** | **66.95** | **91.92** | **86.16** | **90.47** | **85.88** | **88.61** |

Table 1: **Comparison of ToolTree with other baselines across GTA and m&m.** The experiment is carried out under both step-by-step and end-to-end mode. "Tool" stands for tool selection F1 score; "Arg" stands for argument prediction F1 score; "Plan" and "Exec" stand for planning and execution F1 score. Ours achieves the best performance overall.

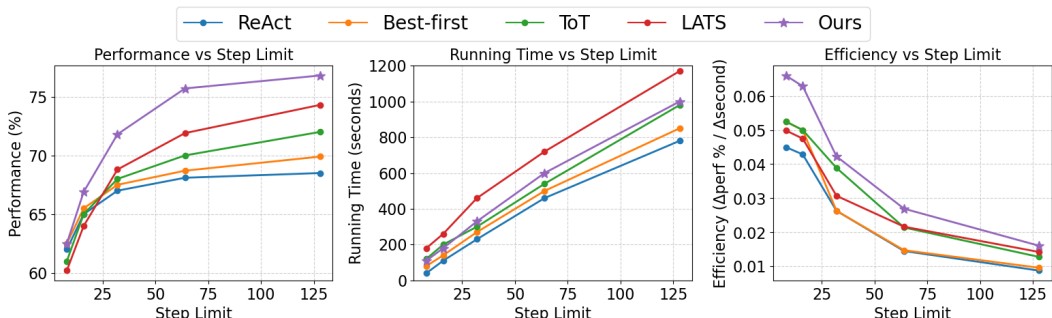

Figure 3: **Progressive efficiency analysis across step limits.** (a) *Performance vs. step limit*; (b) *Runtime vs. step limit*; (c) *Efficiency vs. step limit*. *ToolTree* achieves the highest efficiency compared with baselines. mprovements are largest for step limits between 12 and 64.

## 4.2 CLOSED-SET TOOL PLANNING ON GTA AND M&M

We compare *ToolTree* with the selected baselines on GTA and m&m on both step-by-step and end-to-end mode under GPT-4o and GPT-4o-mini. For step-by-step mode, we measure Tool F1 and Arg F1 to evaluate the tool selection and argument prediction ability. For end-to-end mode, we report F1 score for both planning and execution. GTA and m&m offer typed tool APIs and gold protocols across two modes, allowing us to cleanly measure both planning and execution quality.

**Results.** As demonstrated in Table 1, *ToolTree* attains the best overall average score on both datasets and model backends. On GTA with GPT-4o, it achieves 66.95 average score, outperforming the vanilla MCTS baseline by more than 2.2 points. On m&m with GPT-4o, *ToolTree* reaches 88.61 average score of both modes, outperforming the zero-shot baseline by more than 8 points. The same pattern holds for GPT-4o-mini with smaller but consistent margins. Meanwhile, greedy controllers like Zero-shot, ReAct and CoT lag behind search-based methods, confirming the value of lookahead

| Model | Method | RestBench–TMDB | | | RestBench–Spotify | | | ToolBench | | |
|---|---|---|---|---|---|---|---|---|---|---|
| | | Pass | Win | AVG | Pass | Win | AVG | Pass | Win | AVG |
| GPT-4o-mini | Zero-shot | 33.28 | 50.00 | 41.64 | 26.44 | 50.00 | 38.22 | 28.85 | 50.00 | 39.42 |
| | CoT | 34.42 | 54.70 | 44.56 | 29.82 | 53.10 | 41.46 | 26.29 | 55.47 | 40.88 |
| | ReAct | 38.82 | 61.06 | 49.94 | 32.64 | 59.95 | 46.30 | 34.30 | 58.94 | 46.62 |
| | DFSDT | 46.20 | 64.26 | 55.23 | 35.10 | 65.47 | 50.28 | 38.84 | **68.29** | 53.57 |
| | LATS | 51.33 | 66.67 | 59.00 | 39.81 | **72.85** | 56.33 | 40.08 | 65.77 | 52.92 |
| | **Ours** | **55.17** | **70.40** | **62.79** | **42.08** | 72.18 | **57.74** | **42.24** | 67.90 | **55.07** |
| GPT-4o | Zero-shot | 56.28 | 50.00 | 53.14 | 49.54 | 50.00 | 49.77 | 47.58 | 50.00 | 48.79 |
| | CoT | 58.52 | 52.32 | 55.42 | 47.92 | 44.55 | 46.23 | 46.88 | 47.57 | 47.23 |
| | ReAct | 62.42 | 66.17 | 64.30 | 53.27 | 60.72 | 57.00 | 52.38 | 63.39 | 57.89 |
| | DFSDT | 66.57 | 69.08 | 67.82 | 55.48 | 71.63 | 63.55 | 54.86 | 68.59 | 61.73 |
| | LATS | 68.26 | 74.44 | 71.35 | **61.25** | 75.80 | 68.53 | 59.25 | 73.85 | 66.55 |
| | **Ours** | **72.40** | **75.59** | **74.50** | 60.87 | **78.84** | **71.36** | **61.27** | **76.81** | **69.04** |

Table 2: **Open-set tool-planning results on RestBench and ToolBench using GPT-4o-mini and GPT-4o as back-end LLMs.** Higher values indicate better performance; the best score for each dataset-model pair is highlighted in bold. *"Pass"* and *"Win"* refer to pass rate and win rate.

even with small typed tool suites. Among the rest baselines, while ToT, A* and LATS improve progressively, *ToolTree* remains on top as its *dual pre-/post-evaluation with pruning* filters implausible actions before expansion and cuts unproductive branches after execution using real feedback, concentrating budget on promising chains and yielding higher next-action and executed-plan scores.

**Progressive Efficiency Analysis.** We sweep the step limit and record the dataset performance, wall-clock time and efficiency, defined as the marginal gain per second, at each budget in Figure 3. On Figure *(a) Performance vs step limit*, all methods improve with more steps, but *ToolTree* dominates at every budget, with the largest margin in the low–mid regime of 16–64 steps before all curves begin to saturate, demonstrating the effectiveness of lookahead converting early expansions into higher-quality actions. On Figure *(b) Running time vs step limit*, runtime grows near-linearly for all methods. While *ToolTree* is slower than ReAct and Best-first, it is comparable to ToT and typically below LATS. On Figure *(c) Efficiency vs step limit*, despite the extra time, *ToolTree* yields the highest accuracy-per-second, especially from 16-32 and 32-64 steps, indicating better budget allocation. The pattern aligns with our design: pre-evaluation pruning removes implausible children before expansion and post-evaluation pruning trims unproductive branches after probes, together producing the best performance–time trade-off and a practical sweet spot around 32–64 steps.

## 4.3 Open-set Tool Planning on ToolBench and RestBench

We compare *ToolTree* with Zero-shot, Chain-of-Thought, ReAct, DFSDT, and LATS on ToolBench and RestBench using GPT-4o-mini and GPT-4o. Following each benchmark's protocol, we report Pass Rate and Win Rate under identical instructions, a fixed retrieval setup, and budget parity. These benchmarks expose large, diverse API catalogs and require both API selection and argument composition over executable REST endpoints, providing a clean stress test of planner scalability in many-API, real-world settings.

**Results** As demonstrated in Table 2, *ToolTree* attains the best score across both datasets and models. On ToolBench with GPT-4o, it reaches 69.04 AVG, about +2.5 over the strongest baseline; on RestBench–TMDB, it achieves 74.50 AVG, about +3.1 over the next best. The advantage is largest where branching is high and plans span multiple calls. As our method explores pre-evaluation to filter schema- or slot-incompatible calls before expansion, and applies post-evaluation to prune branches quickly using execution feedback, the resulting value backups favor API sequences that are compatible over longer horizons. In contrast, DFSDT and LATS either allocate depth without breadth-aware priors or distribute rollouts less selectively, leading to inaccurate planning and execution. More results can be found in Appendix A. Potential concerns related to metric coupling are discussed in Appendix A.9.

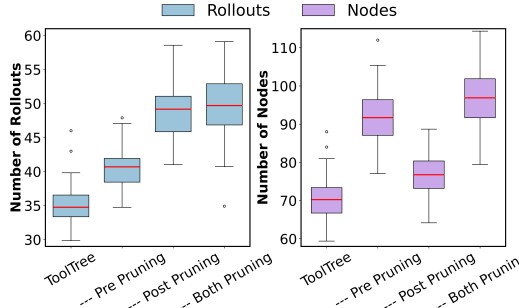

Figure 4: Efficiency comparison of ToolTree and its pruning variants on nodes and rollouts.

| Variant | Accuracy ↑ | Token Cost ↓ |
|---|---|---|
| ToolTree | **76.44** | **18.2k** |
| – Pre-pruning | 75.28 | 20.4k |
| – Pre-evaluation | 71.80 | 21.1k |
| – Post-pruning | 75.82 | 22.9k |
| – Post-evaluation | 68.94 | 22.9k |
| – Both Pruning | 74.58 | 24.1k |
| – Both Evaluation | 66.70 | 24.3k |

Table 3: Ablation of dual evaluation and bidirectional pruning on accuracy and token cost.

**Retrieval Sensitivity**. To isolate the impact of the shortlist, we replace the retriever with Contriever, RoBERTa, and BM25 and evaluate ReAct, ToT, and our planner on ToolBench, reporting the three official instruction groups (G1/G2/G3) as in Table 4. While stronger retrieval lifts all methods, ours remains best across G1–G3 under every retriever. Besides, we also found degradation under weaker retrieval is smallest for our planner, demonstrating the effectiveness of both pre-evaluation and post evaluation on retrieved tool lists. We further attach the result for increasing tool library from 14 to 10014 in the Appendix A.10 to demonstrate its scalability.

| Retriever | Method | G1 | G2 | G3 |
|---|---|---|---|---|
| Contriever | ReAct | 61.0 | 78.0 | 72.5 |
| | ToT | 62.8 | 79.6 | 75.2 |
| | Ours | **64.5** | **81.8** | **78.3** |
| RoBERTa | ReAct | 60.5 | 76.5 | 73.0 |
| | ToT | 63.0 | 80.0 | 76.2 |
| | Ours | **66.0** | **83.0** | **82.8** |
| BM25 | ReAct | 58.2 | 74.1 | 69.4 |
| | ToT | 60.1 | 76.0 | 71.8 |
| | Ours | **62.4** | **79.0** | **74.2** |

Table 4: Retriever ablation on ToolBench.

## 5 ANALYSIS

**Effect of dual evaluation and pruning.** We ablate the effectiveness of dual evaluation and pruning under the same step limits and prompts on GTA with GPT-4o. Table 3 shows that *ToolTree* attains the highest accuracy at the lowest token cost. Removing post-evaluation causes the largest accuracy drop by more than 7 points, indicating that shallow execution feedback is critical for steering search. Concurrently as demonstrated in Figure 4, removing pre-pruning substantially reduces the median number of nodes expanded to approximately 70 from 95 by directly curtailing unpromising branch explorations for a narrower search tree.

Conversely, removing post-evaluation pruning more substantially reduces median rollouts to approximately 33 from 47, as its accurate rewards provide clearer solution quality signals for potentially earlier confident convergence. We provide further analysis on the robustness of using LLM judge for dual evaluation and pruning as illustrated in Appendix A.6.

**Effect of Model Size on Planning.** We study how backbone capacity interacts with our method by sweeping two open-source model families, LLaMA and Qwen, over increasing sizes on GTA and ToolBench. It can be seen in Figure 5 that performance scales monotonically with size for both families on both datasets, with the steepest gains from small to mid models and di-

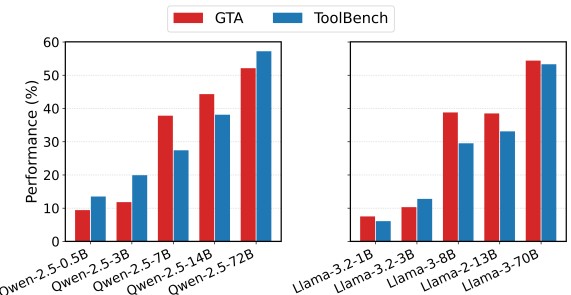

Figure 5: Analysis of Performance with respect to model size on Qwen and LLaMA family.

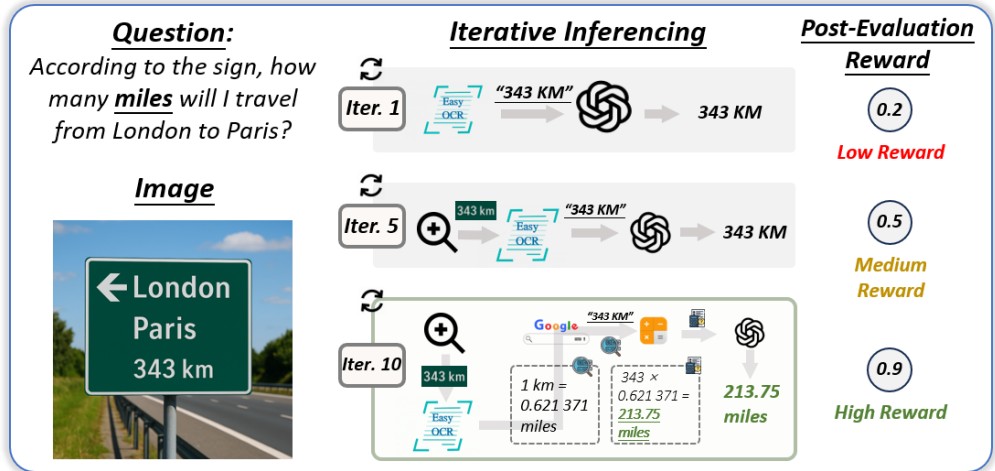

Figure 6: A Sample Case of ToolTree on GTA.

minishing returns thereafter. Besides, we found that ToolBench is more size-sensitive than GTA as larger models help more when the planner must select among many APIs and ground longer argument strings.

**Case Study.** Figure 6 showcases how ToolTree progressively corrects itself on a GTA task. With the number of rollouts grows, ToolTree finds better tool trajectories guided by both the pre-evaluation score as the prior and the post-evaluation score as the dominant reward. The query asks, "According to the sign, how many *miles* is it from London to Paris?"; the photo shows "343 km." In its first rollout, the agent invokes a lightweight OCR tool, passes the raw text to the LLM, and naively returns "343 km," earning a low post-evaluation score (0.2). By the fifth rollout, the search has inserted the *patch-zoom* tool to crop the numeric region and rerun OCR, but it still reports kilometers and receives only a medium reward (0.5). Guided by these signals, the tenth rollout adds a unit-conversion API after OCR; the calculator multiplies $343 \times 0.621\ 371$, and the LLM outputs the correct "213.75 miles," which the judge scores 0.9. More case studies are in Appendix A.8.

# 6 RELATED WORK

**Tool Planning for LLM Agents.** Dynamic tool planning is crucial for complex tasks that require the use of sequential tools (Qu et al., 2025). In order to mitigate such a problem, prompt-based methods leverage LLMs with their strong world knowledge priors (Hao et al., 2023; Gu et al., 2024) as a planner to select tools using in-content learning techniques, such as chain-of-thought (Wei et al., 2022) or ReAct (Yao et al., 2023b) schema (Shen et al., 2023; Paranjape et al., 2023; Lu et al., 2025). Even though flexible, these approaches often make greedy, single-step choices without adequate looking-ahead or backtracking, potentially leading to hallucinated or incorrect actions (Qin et al., 2023; Liu et al., 2024; Zhang et al., 2025). Alternatively, training-based methods fine-tune models or add specific heads for tool invocation (Schick et al., 2023; Yang et al., 2023), incurring significant computational and data annotation costs. ToolTree departs fundamentally from linear pipelines by integrating the Pre-Evaluation score ($r_{\text{pre}}$) directly into the UCT formula to dynamically steer exploration, while the Post-Evaluation score ($r_{\text{post}}$) governs Backpropagation, together forming a non-linear, self-correcting decision policy.

**Augmenting LLM Agent with Tree Search.** To address the limitations of reactive LLM agents in complex tasks requiring lookahead (Gu et al., 2024), augmenting them with tree search provides a deliberate planning layer. Various search algorithms, such as greedy search (Yao et al., 2023b), A* Search (Zhuang et al., 2024), Beam Search (Xie et al., 2023), MCTS (Zhou et al., 2024; Hao et al., 2023), BFS/DFS (Yao et al., 2023a), Best-first search (Koh et al., 2024) have been integrated at inference time. However, these methods often lack sufficient tool invocation diversity for broad domain generalization. Our approach addresses this with explicit tree search for tool selection, contrasting with LLM-internal reasoning prevalent in prior methods, and further incorporates dual environmen-

tal feedback for robust verification and plan refinement. One notable related work with ToolTree is Toolchain* (Zhuang et al., 2024), we attach more comparisons with Toolchain* in Appendix B.3. .

## 7 CONCLUSION

This paper presents ToolTree, a training-free agent framework that integrates a plug-and-play MCTS-based tool planning module to enable robust multi-tool orchestration across diverse tasks. ToolTree explores a dual feedback mechanism from the environment to provide nuanced guidance for MCTS, enabling both efficient search via strategic pruning and effective discovery of optimal tool trajectory. Experiments over 4 datasets across diverse domains of both closed-set and open-set tool planning demonstrate ToolTree consistently outperforms state-of-the-art planning paradigm by 10 percent on average success rate. We hope this method will serve as a valuable foundation for future explorations into sophisticated tool orchestration and reasoning in more advanced AI agents.

**Ethics Statement.**   We affirm adherence to the ICLR Code of Ethics. Our study uses only public benchmarks (GTA, m&m, ToolBench, RestBench) without human subjects or personally identifiable data. API interactions are restricted to benchmark-provided virtual endpoints or public test servers; no private user data or production systems are accessed. Potential risks include automation bias, unintended amplification of model biases, and misuse of automated tool-calling; to mitigate this, we (i) pin evaluator versions and judge prompts, (ii) report multiple seeds and confidence intervals to avoid cherry-picking, (iii) release prompts/tool specs for external auditing, and (iv) follow dataset and API licenses/ToS. We report compute budgets and runtime to encourage awareness of environmental cost. There are no conflicts of interest or external sponsorship that would bias work.

**Reproducibility Statement.**   We provide a complete specification of the problem setup and notation in §**Preliminaries** and the full algorithm (scoring, selection, widening, pruning, backups) with pseudocode and hyperparameters in §**Method**. Experimental protocols (datasets, metrics, budgets, baselines), prompts/tool cards, retrieval settings, and evaluator configurations are detailed in §**Experiments** and the Appendix. We will release an anonymized repository containing code, prompts, tool specifications, evaluation scripts (including judge prompts/versions), seed control, and config files.

**Acknowledgements**   This research was supported by the Korea Planning & Evaluation Institute of Industrial Technology (KEIT) funded by the Ministry of Trade, Industry and Energy (No.RS-2025-25458052, Development of Core Technologies for Manufacturing Foundation Models) and the Institute of Information & communications Technology Planning & Evaluation (IITP) grant funded by the Korea government(MSIT) (No.RS-2025-02217259, Development of self-evolving AI bias detection-correctionexplain platform based on international multidisciplinary governance)

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

| Feature | Ours | General LLM Agent Framework | | Tool Augmented LLM System | | | LLM Agent Tree Search | | |
|---|---|---|---|---|---|---|---|---|---|
| | | GPT-Functions (OpenAI, 2024) | OctoTools (Lu et al., 2025) | HuggingGPT (Shen et al., 2023) | ToolChain* (Zhuang et al., 2024) | ToolPlanner (Liu et al., 2025) | ReAct (Yao et al., 2023b) | Reflexion (Shinn et al., 2023) | LATS (Zhou et al., 2024) |
| Tool Calling | ✓ | ✓ | ✓ | ✓ | ✓ | ✓ | ✓ | ✓ | ✓ |
| Planning | ✓ | ✓ | ✓ | ✗ | ✓ | ✓ | ✓ | ✓ | ✓ |
| Deliberate Tool Selection | ✓ | ✗ | ✗ | ✗ | ✓ | ✓ | ✗ | ✗ | ✓ |
| Tool Verification | ✓ | ✗ | ✓ | ✗ | ✓ | ✓ | ✗ | ✗ | ✓ |
| Tool Refinement | ✓ | ✗ | ✓ | ✗ | ✓ | ✓ | ✗ | ✓ | ✗ |
| Tool Pruning | ✓ | ✗ | ✗ | ✗ | ✗ | ✗ | ✗ | ✗ | ✓ |

Table 5: A comparison of **ToolTree** with notable LLM agent frameworks, tool-augmented LLM systems and LLM agent tree search. Our method shows significant advantages in tool integration.

Andy Zhou, Kai Yan, Michal Shlapentokh-Rothman, Haohan Wang, and Yu-Xiong Wang. Language agent tree search unifies reasoning, acting, and planning in language models. In Ruslan Salakhutdinov, Zico Kolter, Katherine Heller, Adrian Weller, Nuria Oliver, Jonathan Scarlett, and Felix Berkenkamp (eds.), *Proceedings of the 41st International Conference on Machine Learning*, volume 235 of *Proceedings of Machine Learning Research*, pp. 62138–62160. PMLR, 21–27 Jul 2024. URL https://proceedings.mlr.press/v235/zhou24r.html.

Shuyan Zhou, Frank F Xu, Hao Zhu, Xuhui Zhou, Robert Lo, Abishek Sridhar, Xianyi Cheng, Tianyue Ou, Yonatan Bisk, Daniel Fried, et al. Webarena: A realistic web environment for building autonomous agents. *arXiv preprint arXiv:2307.13854*, 2023.

Yuchen Zhuang, Xiang Chen, Tong Yu, Saayan Mitra, Victor Bursztyn, Ryan A. Rossi, Somdeb Sarkhel, and Chao Zhang. Toolchain*: Efficient action space navigation in large language models with a* search. In *The Twelfth International Conference on Learning Representations*, 2024. URL https://openreview.net/forum?id=B6pQxqUcT8.

# A ADDITIONAL EXPERIMENT RESULTS

## A.1 RESULTS ON AGENT FRAMEWORKS

We evaluate ToolTree against three distinct multi-tool orchestration baselines: Few-Shot prompting, HuggingGPT, and OctoTools with two backbone models, GPT-4o-mini and GPT-4o. As covered in Table 6, our evaluation spans 15 datasets in five diverse domains, including general visual, medical, external knowledge, math, and text/document.

| Domain | Dataset | GPT-4o-mini | | | | GPT-4o | | | |
|---|---|---|---|---|---|---|---|---|---|
| | | Few-Shot | HuggingGPT | OctoTools | ToolTree (Ours) | Few-Shot | HuggingGPT | OctoTools | ToolTree (Ours) |
| General Visual | VQAv2 | 68.82 | 60.17 | 69.28 | **74.47** | 73.22 | 67.77 | 74.18 | **76.43** |
| | GQA | 63.80 | 65.13 | 66.14 | **71.54** | 66.84 | 60.33 | 68.58 | **74.44** |
| | SQA | 76.50 | 70.82 | 78.29 | **84.28** | 82.15 | 78.45 | 84.13 | **87.33** |
| Medical | MedQA | 79.14 | 84.33 | 86.18 | **91.13** | 83.20 | 86.73 | 92.17 | **93.88** |
| | VQA-Rad | 48.10 | 55.14 | 60.10 | **63.27** | 54.47 | 58.88 | 66.42 | **74.12** |
| | PathVQA | 24.90 | 40.72 | 43.13 | **47.12** | 26.20 | 37.82 | 46.17 | **50.86** |
| External Knowledge | OK-VQA | 48.46 | 44.19 | 50.17 | **55.38** | 53.62 | 50.12 | 53.42 | **59.27** |
| | A-OKVQA | 60.28 | 55.81 | 62.15 | **70.54** | 65.91 | 60.33 | 68.33 | **73.48** |
| | WebQ | 50.20 | 56.24 | 61.12 | **64.28** | 56.41 | 58.18 | 63.44 | **67.94** |
| Math | MATH | 53.26 | 45.14 | 58.43 | **69.42** | 61.45 | 53.51 | 68.57 | **78.19** |
| | Game-24 | 26.50 | 22.66 | 34.18 | **43.33** | 33.15 | 25.43 | 40.18 | **47.85** |
| | MathVista | 52.53 | 55.62 | 57.97 | **63.14** | 59.10 | 58.44 | 61.70 | **65.58** |
| Text / Doc. | TextVQA | 72.42 | 68.24 | 74.69 | **82.26** | 76.28 | 70.14 | 77.17 | **85.43** |
| | Doc-VQA | 83.28 | 83.10 | 84.23 | **89.43** | 87.11 | 82.13 | 89.39 | **92.33** |
| | HotpotQA | 37.29 | 46.48 | 48.11 | **54.15** | 43.77 | 51.82 | 53.14 | **56.33** |
| | **Average** | 56.37 | 56.92 | 62.94 | **68.65** | 61.53 | 60.01 | 67.80 | **72.70** |

Table 6: Comparison across 15 datasets in five domains. ToolTree consistently outperforms standard few-shot prompting, HuggingGPT, and OctoTools on both GPT-4o-mini and GPT-4o, achieving highest overall score.

Our framework consistently achieves superior performance across five domains. Under GPT-4o-mini, it attains an average of 68.65%, outperforming Few-Shot and HuggingGPT by over 11.7 points

| Configuration | VQA-Rad | OK-VQA | MathVista | SQA | HotpotQA | AVG |
|---|---|---|---|---|---|---|
| **LangChain** | 62.18 | 49.18 | 54.24 | 76.59 | 39.82 | 56.40 |
| w/o COT-SC | 65.14 | 54.37 | 56.88 | 82.17 | 44.90 | 60.69 |
| w/o ReAct | 66.28 | 52.33 | 59.25 | 80.95 | 45.18 | 60.80 |
| w/o ToT | 63.04 | 48.12 | 65.33 | 78.33 | **52.28** | 61.42 |
| **w/o ToolTree** | **67.72** | **54.27** | **65.74** | **81.33** | 51.94 | **64.20** |
| **MetaGPT** | 64.13 | 53.84 | 54.88 | 78.16 | 37.72 | 57.75 |
| w/o COT-SC | 68.74 | 54.88 | 60.30 | 79.44 | 46.90 | 62.05 |
| w/o ReAct | 66.32 | 55.11 | 58.94 | 80.54 | 49.56 | 62.09 |
| w/o ToT | 65.42 | 50.52 | 60.14 | 80.47 | **56.21** | 62.55 |
| **w/o ToolTree** | **69.24** | **55.83** | **62.28** | **82.57** | 54.77 | **64.94** |

Table 7: Comparison of ToolTree as a plug-and-play module with ReAct, COT-SC and ToT modules. ToolTree achieves highest score on average.

and OctoTools by 5.71 points on average. A similar trend is observed with the more capable GPT-4o backbone, where ToolTree outperforms Few-Shot and HuggingGPT by more than 11.1 points and OctoTools by 4.9 points on average. Notably, ToolTree demonstrates substantial gains on traditionally challenging, domain-specific datasets such as PathVQA and Game-of-24, with 22.22% and 16.83% performance gain compared with few-shot baselines under GPT-4o-mini. These significant improvements underscore the superiority of our framework that integrates a domain specialized tool library and MCTS-based tool selector.

## A.2 PLUG-AND-PLAY MODULE COMPARISON

We evaluated our plug-and-play module ToolTree-Module on one representative dataset from each of the five domains under two off-the-shelf LLM-agent frameworks, LangChain and MetaGPT. For each framework, we start from the vanilla agent with no extra tool use module and then insert exactly one of four modules—Chain-of-Thought Self-Consistency (COT-SC), ReAct, Tree-of-Thought (ToT), or our proposed ToolTree-Module—while holding all other settings like prompt format, tool APIs, number of iterations/trajectories, and random seeds identical.

As Table 7 shows, our ToolTree-Module consistently achieves the highest overall average accuracy and outperforms all baselines on four out of five benchmarks across both frameworks, outperforming COT-SC, ReAct, and ToT by 3–8 points on each dataset and 7 points on average against the unaugmented agent. The only exception is HotpotQA, where tree-of-thought's structured reasoning over LLM's hidden state excels at systematically decomposing the multi-hop problem and exploring diverse evidence-linking pathways crucial for this dataset. Nevertheless, this internal state search nature also makes it far worse than our module in domain-specialized tasks that require external tools such as vision, medical and knowledge, where our module's versatile integration and adaptive orchestration of these tools yields significantly better performance.

## A.3 PERFORMANCE COMPARISON FOR EACH DOMAIN

Figure 7 presents a detailed breakdown comparison of ToolTree's average performance across five specialized domains under two backbone models. ToolTree consistently achieves the highest performance across all domains, particularly excelling in the Math and External Knowledge domains. For example, in the Math domain under GPT-4o-mini, ToolTree reaches 63.4%, significantly outperforming Few-Shot by 19.3%, HuggingGPT by 22.2%, and OctoTools by 11.2%. Similarly notable gains are observed under GPT-4o.

In the Medical domain, ToolTree surpasses HuggingGPT by 12.0% and OctoTools by 5.0% using GPT-4o-mini, demonstrating its strength in tasks requiring specialized external knowledge and precise tool interactions. In the General Visual and Text/Document domains, ToolTree continues to show consistent improvements of roughly 5 − 7% over baselines for both backbone models. These results underscore the robustness of ToolTree's MCTS-based tool selection and dual evaluation across diverse reasoning challenges.

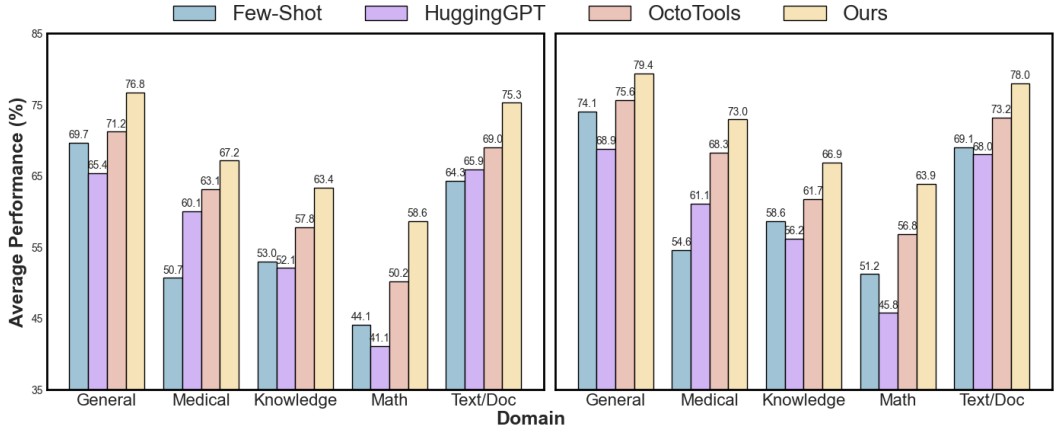

Figure 7: Breakdown comparison for each of the domains.

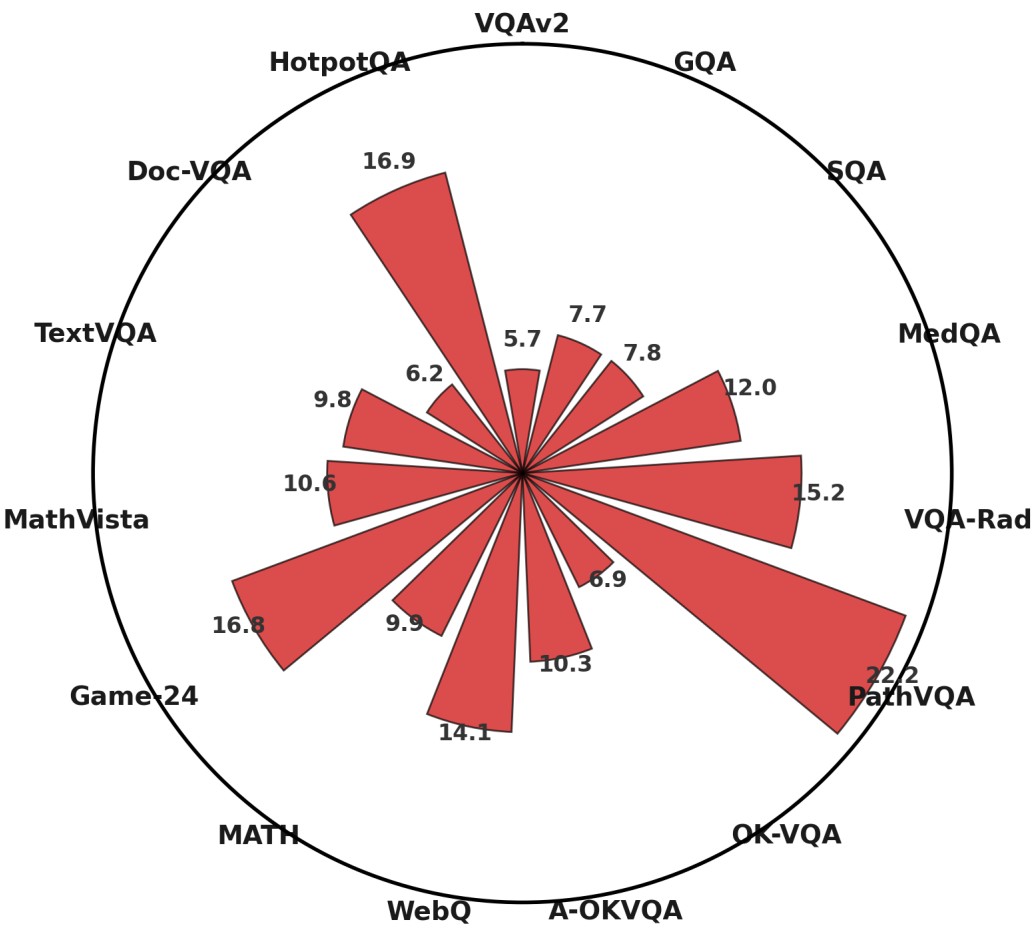

Figure 8: Breakdown comparison with the few-shot baseline setup under GPT-4o-mini.

## A.4 Performance Comparison with Baseline

We measured ToolTree's per-dataset improvement over a GPT-4o few-shot baseline by subtracting the baseline accuracy from ToolTree's accuracy on each of the fifteen tasks and plotting the results in Figure 8. The chart shows gains on every benchmark: PathVQA sits at the top with an uplift

exceeding twenty points, followed by the Game of 24 and HotpotQA climbing into the mid-teens, and VQA-Rad and A-OKVQA rising by around 15% and 10% respectively. Even general visual tasks like VQAv2 and TextVQA register solid improvements of roughly six to eight points. This pattern reflects ToolTree's strength in orchestrating multi-step, domain-specialized tool chains that is essential for medical and mathematical puzzles, while its verification and pruning mechanisms consistently enhance performance on more conventional downstream tasks.

## A.5   RESULTS ON APIBENCH

We further carried out additional results on APIbench to demonstrate its applicability in tool the invocation task as illustrated in Table 8. This confirms that our dual evaluation mechanism is not merely a pipeline heuristic but a generalized hallucination filter. It successfully identifies and prunes invalid tool candidates in zero-shot settings on completely unseen libraries, without the need for domain-specific fine-tuning

Table 8: APIBench results (BM25 retriever) for GPT-4o and GPT-4o-mini with different planning strategies. We report AST-based overall accuracy (%) on HuggingFace, TensorHub, and TorchHub, as well as the macro-average accuracy and hallucination rate across the three subsets.

| Backbone | Method | HuggingFace | TensorHub | TorchHub | Avg. Acc. (%) | Avg. Hallu. (%) |
|---|---|---|---|---|---|---|
| GPT-4o-mini | Zero-shot | 68.4 | 59.2 | 44.5 | 57.4 | 22.1 |
| | ReAct | 69.8 | 61.5 | 47.2 | 59.5 | 18.5 |
| | Tree-of-Thought | 71.2 | 62.8 | 49.6 | 61.2 | 9.3 |
| | ToolTree (Ours) | 73.5 | 65.4 | 53.1 | 64.0 | 7.4 |
| GPT-4o | Zero-shot | 76.5 | 69.8 | 62.3 | 69.5 | 7.8 |
| | ReAct | 77.2 | 71.0 | 63.5 | 70.6 | 5.1 |
| | Tree-of-Thought | 78.0 | 72.4 | 64.8 | 71.7 | 2.5 |
| | ToolTree (Ours) | 79.2 | 74.1 | 66.5 | 73.3 | 2.1 |

## A.6   ROBUSTNESS TO LLM-AS-JUDGE NOISE.

A potential vulnerability of ToolTree is its reliance on LLM-based judgment for pre- and post-evaluation. To quantify this risk, we conduct a *restoration analysis* on ToolBench, where we start from actual ToolTree trajectories and counterfactually correct erroneous judge decisions on a random subset of instances. We consider three variants: (i) selectively fixing false positives (rejecting tool calls that the judge incorrectly approved), (ii) selectively fixing false negatives (accepting tool calls that the judge incorrectly rejected), and (iii) an oracle setting where all judge decisions are corrected. Table 9 reports the judge error rate and task success rate for GPT-4o and GPT-4o-mini under these configurations.

| Backbone | Configuration | Judge error rate | Task success (%, $\Delta$) |
|---|---|---|---|
| GPT-4o | ToolTree (baseline) | 25.8% | 51.9%   (—) |
| | + Fix false positives | 7.4% | 52.5%   (+0.6) |
| | + Fix false negatives | 18.4% | 54.1%   (+2.2) |
| | Oracle (perfect judge) | 0.0% | 54.7%   (+2.8) |
| GPT-4o-mini | ToolTree (baseline) | 39.4% | 49.5%   (—) |
| | + Fix false positives | 16.3% | 50.8%   (+1.3) |
| | + Fix false negatives | 23.1% | 52.4%   (+2.9) |
| | Oracle (perfect judge) | 0.0% | 53.6%   (+4.1) |

Table 9: Restoration analysis of LLM-judge errors on ToolBench. "Judge error rate" is the fraction of incorrect pre-/post-evaluation decisions. "Task success" is the overall pass rate; $\Delta$ denotes the absolute difference (in percentage points) relative to the actual ToolTree baseline for each backbone.

The restoration results yield two main observations. First, ToolTree exhibits *empirical tolerance* to judge noise: despite non-trivial error rates (25.8% for GPT-4o and 39.4% for GPT-4o-mini), the performance gap to an oracle judge is modest (at most +2.8 and +4.1 points, respectively). More-over, correcting only false positives yields limited gains, while correcting false negatives accounts

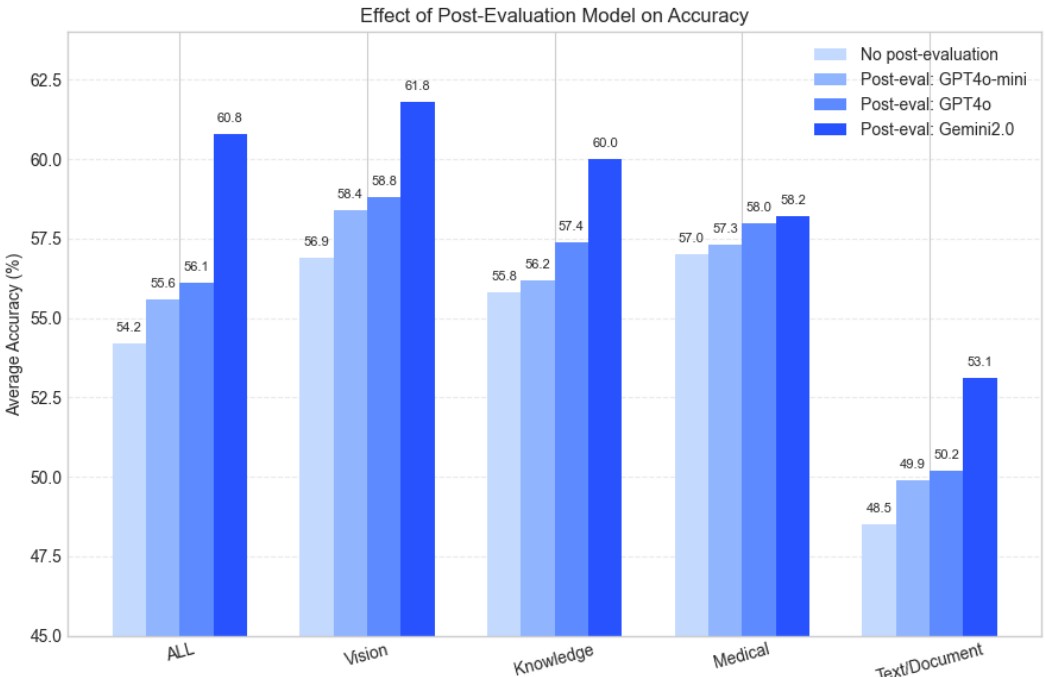

Figure 9: Effect of different LLM for post evaluation.

| Internal judge | Benchmark evaluator | ToolBench | RestBench |
|---|---|---|---|
| GPT-4o | GPT-4o | 69.04 | 72.48 |
| Gemini-2.5-Flash | GPT-4o | 72.71 | 73.12 |
| LLaMA-3.3-70B | GPT-4o | 46.48 | 50.17 |
| LLaMA-3.3-70B | LLaMA-3.3-70B | 38.11 | 41.64 |

Table 10: Cross-vendor / cross-judge robustness of ToolTree on ToolBench and RestBench (pass rate %). We vary the internal judge used during planning and the external benchmark evaluator.

for most of the improvement, indicating that overly conservative judgments are more harmful than permissive ones. Second, the search does not collapse under noisy judgments because the LLM signals enter the planner as *soft guidance* rather than ground truth: $r_{pre}$ and $r_{post}$ are bounded priors inside the MCTS update, and their influence is aggregated over many rollouts. ToolTree repeatedly revisits and reevaluates actions, so isolated misjudgments are statistically smoothed out instead of being irrevocably baked into a single greedy trajectory.

## A.7 Effect of Dual Feedback on Accuracy

We measured how the choice of post-evaluation model affects overall and domain-specific accuracy by running the MCTS pipeline under four settings: no post-evaluation, GPT-4o-mini as judge, GPT-4o as judge, and Gemini 2.0 as judge, as shown in Figure 9. Across all tasks, accuracy steadily increases with more powerful judges, rising from 54.2% to 60.8% (Gemini 2.0). The largest improvements appear in vision and text/document tasks, where nuanced output verification matters most. These results show that richer post-execution feedback enables the agent to better discriminate useful tool calls, leading to more accurate final answers.

## A.8 Additional Case Study

We show more cases of ToolTree on the evaluation benchmark as shown in Figure 10

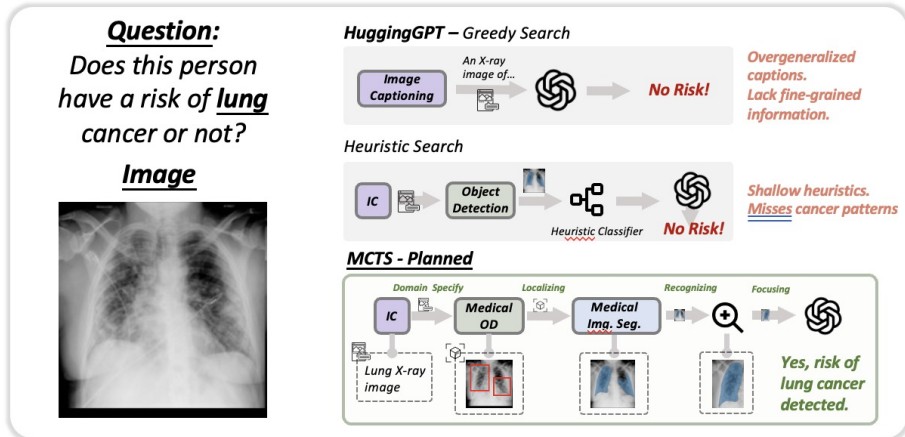

(a) Example inference trajectory for a medical VQA query.

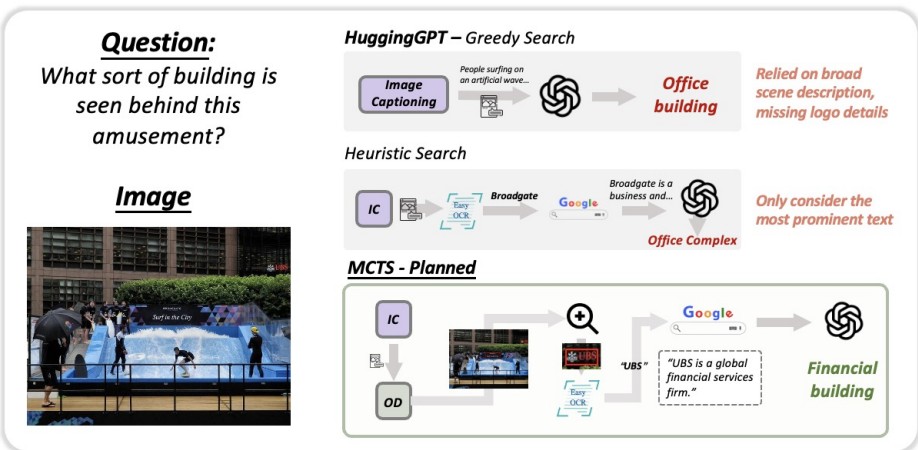

(b) Example inference trajectory for a multi-hop reasoning query.

Figure 10: Two qualitative case studies showcasing ToolTree's iterative tool orchestration on (a) a radiology image question and (b) a multi-hop knowledge reasoning task.

## A.9 CROSS-VENDOR CHECK ON POTENTIAL METRIC COUPLING

A potential concern is that ToolTree might overfit to the particular LLM judge used during planning, especially when the same model family is also used as the benchmark evaluator. To test this, we conduct a cross-vendor ablation where we vary the internal judge used by ToolTree (GPT-4o, Gemini-2.5-Flash, LLaMA-3.3-70B) and the external evaluator (GPT-4o vs LLaMA-3.3-70B) on ToolBench and RestBench. As shown in Table 10, decoupling the judge from the evaluator can improve performance. In contrast, tightly coupling LLaMA-3.3-70B as both judge and evaluator degrades pass rate. These patterns suggest that ToolTree primarily benefits from higher-quality reasoning signals rather than exploiting a specific evaluator.

## A.10 EFFECT OF TOOL LIBRARY SIZE.

While our ToolBench experiments (over 16k tools) already demonstrate large-scale capability, we also conduct a controlled stress test to directly examine ToolTree's robustness to growing tool libraries and noisy tools. Starting from a small closed-set configuration with 14 task-relevant tools, we progressively inject distractor tools drawn from unrelated domains, increasing the total library

| Added distractors | Total tools | Selection strategy | Avg. F1 (%) | Rel. drop |
|---|---|---|---|---|
| 0 (baseline) | 14 | Direct context | 55.89 | / |
| +10 | 24 | Direct context | 55.74 | −0.15% |
| +100 | 114 | Retrieval (Top-20) | 55.15 | −0.74% |
| +1,000 | 1,014 | Retrieval (Top-20) | 54.82 | −1.07% |
| +10,000 | 10,014 | Retrieval (Top-20) | 54.27 | −1.62% |

Table 11: Stress test on tool library size. We start from a 14-tool baseline and progressively add distractor tools. "Avg. F1" denotes average task performance, and "Rel. drop" is the relative performance decrease compared to the 14-tool baseline.

size up to 10,014 tools. For each setting, we report the average F1 score and the relative performance drop compared to the 14-tool baseline. As shown in Table 11, even when the library size increases by three orders of magnitude, the performance degradation remains below 2%, indicating that ToolTree's pre-evaluation module effectively filters out irrelevant tools from a large pool. As the pre-evaluator scores tools based on semantic relevance rather than raw frequency, we can fix the pruning thresholds as it continue to function reliably across all scales. This bridges the gap between our small closed-set benchmarks (e.g., GTA) and large open-set scenarios (e.g., ToolBench), and supports ToolTree's viability for massive, real-world tool libraries.

# B  EXPERIMENT DETAILS

## B.1  BENCHMARK DATASET

We provide comprehensive descriptions of the datasets and baseline methods used in the main experiments. This appendix serves as a reference to understand the task setups, evaluation modes, and implementation details for reproducibility and further analysis.

### B.1  CLOSED-SET TOOL PLANNING BENCHMARKS

**GTA (General Tool Agent)** (Wang et al., 2024). GTA is a benchmark designed to evaluate general-purpose tool use in LLM agents. It defines a fixed set of 14 APIs with well-typed input/output schemas and multi-hop compositional tasks. Each task requires the agent to invoke a series of tools in a logical order to complete the goal. GTA is evaluated under two modes:

- **Step-by-step mode:** Agents plan tool usage iteratively, predicting both the tool and its arguments at each step.
- **End-to-end mode:** Agents must generate the full tool call sequence in a single pass.

**m&m (Multi-modal and Multi-step Tool Use)** (Ma et al., 2024). The m&m dataset features 33 APIs spanning vision, text, and arithmetic tasks. Each task involves integrating multiple modalities (e.g., images, structured text) and planning tool usage over longer horizons. The benchmark emphasizes input schema matching and argument consistency in tool sequences.

### B.2  OPEN-SET TOOL PLANNING BENCHMARKS

**ToolBench** (Qin et al., 2023). ToolBench is a large-scale benchmark that focuses on open-set tool planning with real-world APIs. It consists of 16,464 APIs extracted from online documentation. Each task includes a natural language query, and the agent must 1) retrieve relevant APIs from the entire pool (tool retrieval); 2) generate valid input arguments; and 3) compose executable tool sequences to solve the task. Evaluation follows a judge-based protocol with **Pass Rate** (correct solution) and **Win Rate** (head-to-head comparison against baselines).

**RestBench** (Song et al., 2023). RestBench evaluates agent performance over RESTful APIs in two domains: TMDB (movie database) and Spotify. Unlike ToolBench, the API pool is smaller (143 endpoints), but tasks still require multi-step planning, slot filling, and reasoning over endpoint chains. Evaluation is similar to ToolBench.

## B.3 Baseline Methods

**Zero-Shot.** A vanilla LLM is prompted directly with the task instruction and available tools, without additional planning or prompting heuristics. This serves as a lower-bound baseline.

**ReAct** (Yao et al., 2023b). This method combines reasoning (chain-of-thought) and acting (tool invocation) in an interleaved fashion. At each step, the model generates intermediate reasoning followed by tool calls. It is greedy and reactive, without explicit planning.

**Chain-of-Thought (CoT)** (Wei et al., 2022). CoT decomposes the task via intermediate reasoning steps, but without tool calls. In the context of tool planning, it is extended to select tools after each reasoning span.

**Best-First Search** (Koh et al., 2024). A tree-based search method that prioritizes the expansion of most promising partial plans using heuristics. It expands the most likely paths but does not account for long-term reward or rollout diversity.

**Tree-of-Thought (ToT)** (Yao et al., 2023a). A general planning paradigm where LLM-generated thoughts are expanded into a search tree, with scoring used to backtrack and explore multiple options. ToT treats internal LLM reasoning as planning units, not actual tool execution.

**A\* Search** (Zhuang et al., 2024). Adapts A\* to the tool planning problem by defining a heuristic function over action sequences. It explores the search space by balancing cost so far and expected utility, assuming an accurate reward heuristic. ToolTree is conceptually related to ToolChain\*, which applies A\* search over tool sequences guided by a single heuristic score computed before execution. However, A\* search commits to a best-first expansion based on this heuristic and cannot easily revise early decisions if the heuristic is misaligned with actual tool behavior. In contrast, ToolTree uses MCTS to repeatedly sample and update action values, which allows recovery from early mistakes. Moreover, ToolTree explicitly separates a prior score $r_{pre}$ (used for pre-pruning and exploration) from a grounded post-execution reward $r_{post}$ (used for backpropagation and post-pruning), enabling the planner to discard branches that appear promising in theory but fail in practice. Finally, instead of relying only on queue ordering, ToolTree performs explicit bidirectional pruning before and after tool calls to reduce error propagation and tool cost.

**LATS (Language Agent Tree Search)** (Zhou et al., 2024). LATS is a framework that combines tree search with tool execution, using LLM-guided rollouts and post-hoc scoring to prune weak branches. Unlike ToolTree, it lacks pre-evaluation before tool execution.

**DFSDT** (Qin et al., 2023). This is a strong open-set baseline designed for ToolBench, where the agent uses a depth-first symbolic planner over retrieved APIs, guided by LLM scoring. It focuses on execution consistency rather than search efficiency.

## B.4 Model and Evaluation Protocol

All baselines are implemented under the same backbone model settings (GPT-4o and GPT-4o-mini). For closed-set tasks, the tool APIs are pre-defined and shared with all models. For open-set tasks, we use fixed Top-K API retrieval (K=20) and identical prompt formats. We report:

- **Closed-set:** Tool F1, Argument F1, Plan F1, and Execution F1.
- **Open-set:** Pass Rate and Win Rate, averaged over three instruction templates.

**Hyperparameter Setting.** Evoked by Zhou et al. (2024), during MCTS we set the exploration constant to $\lambda = 1.4$, allow at most $R_{\max} = 60$ roll-outs, and prune branches whenever $r_{\text{pre}} < 0.3$ or $r_{\text{post}} < 0.4$; search stops early if the best $Q$ value increases by $< 10^{-3}$ over 10 consecutive roll-outs.

## B.5 Tool Library

Table **??** summarizes the external tools and models integrated within the ToolTree library, categorized by their domain specialization. The library offers broad coverage across general visual understanding, knowledge-based VQA, medical QA, mathematical reasoning, and text/document tasks. For each domain, a diverse set of functions, ranging from object detection and image segmentation

to knowledge graph querying, medical report generation, and OCR, is supported by state-of-the-art models and APIs. This comprehensive and modular toolset enables ToolTree to handle a wide spectrum of complex, multi-modal tasks with domain-adaptive precision.

### B.6    TOOL CARD METADATA EXAMPLE

We hereby attach the metadata for the medical object detection tool as an illustrative example in Table 12.

| Field | Type | Description / Example |
|-------|------|----------------------|
| `tool_name` | string | `"Medical_Object_Detection"` |
| `description` | string | A tool that detects the organs within a given medical image, such as CT, MRI, X-Ray and pathology images. |
| `input` | `image:  str` | Path to the image file (e.g. `"lung_cancer_Image.png"`) |
| | `prompt:  str` | Prompt to guide detection (default: "Detect the organs in the given image.") |
| `output` | dict | Detected organs with their bounding box, organ name, and confidence score. |
| `example` | `input`
`output` | `{"lung_cancer_Image.png"}`
`{"object_1":  {"name":"left lung", "bounding box":[27,45,31,102], "confidence":0.82},`
`   "object_2":  {"name":"right lung", "bounding box":[57,48,35,98], "confidence":0.82}}` |

Table 12: Metadata schema for the `Medical_Object_Detection` tool.

## B.7  Pre-pruning Judge Prompt for $r_{\text{PRE}}$

---

**System message**

**Role.** You are a strict tool-planning judge for a language-agent that solves user tasks by calling tools in sequence.

**Inputs.** You are given:

- the original user query and current conversation context;
- a tool card (name, description, I/O schema, examples);
- a concrete argument draft that is syntactically valid for the tool.

**Output format.** You must output a single JSON object with:

- `"score"`: a real number between 0.0 and 1.0 (inclusive) measuring how promising this tool call is *before* running it;
- `"explanation"`: a brief natural-language justification (2–4 sentences).

**Scoring guideline.** Use a *coarse* scale in $[0, 1]$. There is no need to finely distinguish every small difference; choose a value that roughly reflects your judgment of usefulness.

**What to penalize.** Give low scores to candidate tool calls that:

- mismatch the required modality or domain;
- ignore key constraints or required fields in the schema;
- duplicate a previous call with effectively identical arguments and no clear new benefit;
- are speculative when a more direct or specific tool is available.

**Important.** Do *not* simulate the tool output; you are judging only the *promised* usefulness of this tool call as the next action.

---

**User message template**

Construct the user message to the judge with the following structure.

**Context.**

- **User query:** USER_QUERY
- **Current dialog / planning context:** CURRENT_CONTEXT

**Candidate tool card.**

- Name: TOOL_NAME
- Description: TOOL_DESCRIPTION
- Input schema: TOOL_INPUT_SCHEMA
- Output schema: TOOL_OUTPUT_SCHEMA
- Example uses (if any): TOOL_EXAMPLES

**Candidate argument draft.**

- Arguments to pass into the tool: ARGUMENT_DRAFT_JSON

Then ask the judge:

*Task: Decide how promising it is to execute this tool call **next** for solving the user's query, given the current state of the conversation and prior tool calls. Please respond **only** with a JSON object of the form* {`"score":  <float between 0.0 and 1.0>, "explanation": "<2--4 sentence explanation>"`}.

---

## B.8 POST-PRUNING JUDGE PROMPT FOR $r_{\text{POST}}$

---

**System message**

**Role.** You are a strict tool-planning judge for a language-agent that solves user tasks by calling tools in sequence.

**Inputs.** You are given:

- the original user query and conversation context *before* the call;
- the tool card;
- the concrete arguments that were used;
- the actual tool output.

**Output format.** You must output a single JSON object with:

- `"score"`: a real number between 0.0 and 1.0 (inclusive) measuring the *grounded utility* of this executed tool call;
- `"explanation"`: a brief natural-language justification (2–4 sentences).

**Scoring guideline.** Use a *coarse* scale in $[0, 1]$. Choose a value that roughly reflects how helpful this call was; you do not need to finely distinguish very small differences.

**When assigning the score, consider:**

- **Task-consistency**: does the output address the user's query or current sub-goal?
- **Correctness / plausibility**: are there obvious errors or contradictions?
- **Relevance**: is the output focused on what is needed now, rather than generic or noisy?
- **Constraint satisfaction**: does it respect safety, formatting, and domain constraints?

**Important.** You are judging only *this* tool call's incremental contribution from the previous context to the new context. Do not re-evaluate the entire plan.

---

**User message template**

Construct the user message to the judge with the following structure.

**Context.**

- **User query:** USER_QUERY
- **Dialog / planning context before this call:** CONTEXT_BEFORE_CALL

**Executed tool card.**

- Name: TOOL_NAME
- Description: TOOL_DESCRIPTION
- Input schema: TOOL_INPUT_SCHEMA
- Output schema: TOOL_OUTPUT_SCHEMA
- Example uses (if any): TOOL_EXAMPLES

**Call details.**

- Arguments actually used: ARGUMENT_JSON
- Tool output (raw): TOOL_OUTPUT_RAW

Then ask the judge:

*Task: Evaluate how much this executed tool call **actually** helped with solving the user's query, considering correctness, relevance, and progress toward a final answer. Please respond **only** with a JSON object of the form* {`"score": <float between 0.0 and 1.0>, "explanation": "<2--4 sentence explanation>"`}.

---

