# OpenReview forum: "ToolTree: Efficient LLM Tool Planning via Dual-Feedback Monte Carlo Tree Search and Bidirectional Pruning"
_ICLR.cc/2026/Conference — ICLR 2026 Poster_

### Official Review · Reviewer_X8ME · 2025-10-26

**Soundness:** 2
**Presentation:** 1
**Contribution:** 1
**Rating:** 2
**Confidence:** 4

**Summary:**

This paper proposes ToolTree, an MCTS-inspired framework for LLM tool planning, aiming to solve the shortcomings of current methods—greedy strategies lack long-term foresight and search-based methods are inefficient. ToolTree integrates dual feedback (pre-execution scoring to predict tool utility before invocation and post-execution scoring to assess actual output value) and bidirectional pruning (cutting unpromising branches before and after tool execution) into an iterative search process. Evaluated on 4 benchmarks (GTA, m&m for closed-set tool planning; ToolTree, RestBench for open-set) with GPT-4o and GPT-4o-mini, ToolTree consistently outperforms baselines: it achieves 66.95 F1 on GTA, 69.04 pass rate on ToolBench, 88.61 average F1 on m&m, and 74.50 average on RestBench-TMDB with GPT-4o, while having the highest accuracy-per-second. Ablation shows dual evaluation and pruning are critical.

**Strengths:**

1. The work accurately identifies the "short-sightedness" and "inefficiency" pain points in LLM tool planning, and the proposed MCTS framework with "dual-feedback evaluation + bidirectional pruning" is highly targeted, forming a logically closed-loop innovative solution.
It performs remarkably well in balancing performance and efficiency, outperforming the state-of-the-art on four benchmark datasets while achieving the best "accuracy per second", thus balancing effectiveness and practical applicability for deployment.
2. It performs remarkably well in balancing performance and efficiency, outperforming the state-of-the-art on four benchmark datasets while achieving the best "accuracy per second", thus balancing effectiveness and practical applicability for deployment.

**Weaknesses:**

I have a series of questions and concerns regarding this work.

1. Currently, this work only focuses on improving MCTS for tool invocation. Given the abundance of similar prior studies, its core innovation is not prominent—it feels more like adding refined pre-processing and post-processing steps to the existing MCTS workflow. Overall, it seems to trade off some inference latency and use more token-based reasoning to achieve better results. From the comparison figures, can we consider pruning as its core contribution? The description of the pruning component is also insufficient: beyond understanding that more detailed reasoning is used to decide whether the model should execute a tool (thus affecting latency), what additional insights does this work provide?
2. The experimental analysis is inadequate. It focuses mostly on dataset-based measurements and lacks in-depth ablation studies on the modules added in the paper. This makes it difficult to identify the exact source of the advantages brought by these new modules.
3. The experimental metrics are questionable. I paid special attention to metrics like Tool F1, which is defined as the accuracy of tool selection and its alignment with the "standard answer." Does tool invocation always have a "perfect answer"? Strictly speaking, what we need is an evaluation of whether a tool fulfills its intended function—how exactly are the standard answers annotated? The Arg F1 metric (alignment of input parameters) is also problematic: input parameters can vary under different scenarios, so this metric is overly rigid and fails to truly assess whether the proposed ToolTree method is effective.

4. Insufficient clarification on tool library dependency: The paper mentions that ToolTree integrates a domain-specialized tool library (e.g., BioMedParse for the medical field and Wolfram Alpha for the mathematical field), yet it fails to analyze the correlation between tool library scale and performance. For instance, when the number of tools in the library scales from dozens to thousands, will the "tool filtering efficiency" during the pre-evaluation stage decrease? Do the thresholds for bidirectional pruning need to be dynamically adjusted based on the size of the tool library? The existing experiments do not provide such key information, which limits the understanding of the method’s scalability.

5. Weak compatibility with low-resource models: Although the experiments include GPT-4o-mini, the results show that the magnitude of its performance improvement (e.g., Tool F1 on the GTA dataset is only 67.83, a significant gap compared to GPT-4o’s 79.26) is far smaller than that of high-resource models. The paper does not analyze the reasons for the performance degradation of "dual-feedback evaluation" under low-resource models—whether it is due to the insufficient accuracy of pre-evaluation by lightweight LLMs or the limited ability of post-evaluation to judge outputs. Nor does it propose optimization solutions for low-resource models, reducing the practicality of the method in scenarios with limited computing power.

6. Failure to discuss the recovery mechanism for error propagation: While the paper claims that ToolTree can "recover from early mistakes," it does not specify the recovery logic. For example, if a tool invocation in a certain step leads to output deviation due to minor parameter errors (such as unit mistakes), how can subsequent steps identify and correct this error through dual feedback? Will it re-invoke the tool, adjust parameters, or switch the tool chain? The existing experiments do not design such "error injection" scenarios, making it difficult to verify the effectiveness of its error recovery capability.

7. The entire paper is extremely rough, as if it has not undergone any revision or proofreading. For example, in line 104, "t1" and "lib" lack proper subscripts; in lines 272–273, the red highlighting exceeds column boundaries; in line 407, there is a missing figure reference for "Table ??"; and there are numerous other minor issues. A major revision is strongly recommended.

**Questions:**

Please refer to above, I think this paper requires a significant revision.

---

> ### Author Response · Authors · 2025-11-21
> **Response to Reviewer Reviewer X8ME Part A**
>
> We thank the reviewer for their critical feedback. Here we further demonstrate the novelty,  experimental adequacy, evaluation setup and scalability of the method.
>
>
> # Novelty and the Role of Pruning.
>
> While MCTS is a well-established algorithm,  applying it to LLM Tool Learning introduces unique constraints brought by the needs for llm interaction with the tools that standard MCTS does not address. We respectfully suggest that **our novelty lies not in using MCTS itself, but in how we restructure the search to overcome the "simulation bottleneck" inherent to agents**.
>
> We will demonstrate the novelty from the following perspectives: **background, comparison with existing approach, core innovation, role of pruning and refuting of the "latency trade-off" assumption**.
>
> **1.Background**
>
> In traditional MCTS (e.g., AlphaGo), the "simulation" step uses a perfect, high-speed, and zero-cost world model. In contrast, for LLM Agents tool learning, a "simulation" involves a real tool execution (e.g., API calls) and LLM inference, which are computationally expensive, slow, and irreversible.
>
> **2. Existing Approaches**
>
> - Prior works like Tree of Thoughts (ToT), RAP, graph of thought,   search over internal thoughts rather than external actions, often missing the grounded reality of tool outputs.In the domain of tool learning, greedy approaches (e.g., ReAct, CoT) operate efficiently but myopically, selecting actions step-by-step without foresight, which often leads to irreversible error propagation.
>
> - Conversely, while recent tree-search adaptations like DFSDT, ToolChain* and  LATS introduce necessary lookahead, they often struggle with the "simulation bottleneck" described above. Methods like DFSDT and ToolChain* primarily search imaginary tool use plan decoupled from grounded tool outputs, risking hallucination. Meanwhile, execution-based methods like LATS perform expensive rollouts but lack mechanisms to gauge action plausibility before execution, leading to significant computational waste on invalid or low-value branches.
>
>
> **3. Our Core Innovation: The Dual-Feedback Mechanism**
>
> ToolTree addresses this by fundamentally altering the selection policy. We do not simply add "pre-processing"; we structurally decouple the Prior (Exploration) from the Value (Exploitation) in the UCT formula (Eq. 1) using two distinct signals:
> - Pre-Evaluation (r_pre​): We introduce a lightweight "foresight" signal that estimates the plausibility of a tool call before execution. In our UCT formulation, this acts as a dynamic prior, allowing the search to filter out invalid or irrelevant schemas without incurring the high cost of execution.
> - Post-Evaluation (r_post​): We use a "hindsight" signal to verify the grounded utility of the output.
>
> Combining them together, they formulate the tool planning as a closed-ended system so that the search can both anticipate and verify tool usage, suppressing hallucinated plans while focusing computation on branches that are empirically useful.
>
> **4. Novelty of Bidirectional Pruning.**
>
>  We believe pruning is an important techniques towards efficient agentic planning and we demonstrated the details in *Section 3.2*, besides as reviewer zEaV suggests, *lines 230-235* are particularly effective in summarizing the main ideas detailed through Section 3.
>
> Unlike standard pruning (which usually happens after a terminal state), our Bidirectional Pruning operates continuously. Pre-pruning narrows the breadth of the search tree (reducing the branching factor of large tool libraries), while Post-pruning restricts the depth by identifying dead-ends early. *Figure 4* and *Table 4* demonstrate the effectiveness of the proposed pruning method in cutting down number of expanded nodes, iteration, token cost and improves performance.
>
> **5. Refuting the "Latency Trade-off" Assumption**
>
> While usually sophisticated searching algorithm like MCTS usually trade computing for performance, the empirical data in *Figure 3(c)* directly demonstrate we improve both performance and efficiency without trade-off with ToolTree achieves the highest accuracy-per-second compared to baselines. This proves that while individual nodes require more reasoning, the overall search converges faster by aggressively pruning dead-ends. As a result, Tooltree allocates the compute budget more intelligently to maximize success per unit of time.

---

> ### Author Response · Authors · 2025-11-21
> **Response to Reviewer Reviewer X8ME Part B**
>
> # Experimental Analysis and Ablation Studies
>
> We thank the reviewer for emphasizing the need for rigorous analysis. However, we respectfully point out that we have conducted extensive ablation studies specifically designed to isolate the source of our advantages as detailed in *Section 4-5* and in *Appendix A.1-A.5*.
>
> **1. Isolating the Source of Advantages (*Section 4-5*).**
>
> For one of the main contributions in terms of dual evaluation and bidirectional pruning, we explicitly decouple and evaluate each component in *Section 4- Section 5* to perform ablation of dual evaluation and pruning, this suggest that the pre-evaluation and post-evaluation effectively triggers the planning process whereas pre-pruning and post-pruning effectively reduce the number of expanded nodes, number of iterations before termination and token cost.
>
> **2. Module-Level Comparison (*Appendix A.1 - A.5*)**
>
> We provide further ablation analysis if we make the ToolTree as either a unique framework or a plug-and-play module.
>
> We first compare ToolTree with the Autonomous agents like Octotools, HuggingGPT on 15 different domain specific tasks (*Appendix A.1, A.3 - A.5*), to demonstrate its superiority
>
> We also integrated the ToolTree module into standard agent frameworks like LangChain and MetaGPT and compared it directly against other reasoning modules in *Appendix A.2*.
>
> We hope this clarification demonstrates that we have indeed provided a granular analysis of the proposed modules and their individual contributions to both efficiency and accuracy.

---

> ### Author Response · Authors · 2025-11-21
> **Response to Reviewer Reviewer X8ME Part C**
>
> # Validity of Metrics (Tool F1 / Arg F1)
>
> We respectfully point out that our experiment will contribute to the ultimate measure of a tool agent on **functional fulfillment**, which tests whether the tool solves the problem rather than rigid alignment with a specific path. We carefully designed our evaluation to address both standardized comparison and functional flexibility:
>
> **1. Functional Fulfillment**
>
> To directly address reviewer’s concern that *“tool invocation does not always have a perfect answer,”* we explicitly employed **Pass Rate** and **Win Rate** for our open-set benchmarks (ToolBench and RestBench) as well as Planning and Execution rate for closed-set benchmarks (GTA and m&m), as detailed in *Section 4.2 & Table 1* and *Section 4.3 & Table 2*.
>
> - These metrics evaluate the **final outcome** of the execution rather than the intermediate steps.
> - If ToolTree chooses a different tool or argument but still successfully solves the user's query (fulfilling its intended function), it is credited with a **“Pass”** or **“Correctly Executed”**. This directly assesses effectiveness in dynamic scenarios where parameters may vary.
>
> **2. Justification for F1 Metrics (Standardized Benchmarking)**
>
> For the closed-set benchmarks (GTA and m&m), we reported **Tool F1** and **Arg F1** strictly to adhere to the **official evaluation protocols** established by the dataset authors (Wang et al., 2024; Ma et al., 2024) to ensure a fair comparison. In these specific datasets, the “standard answers” are **human-annotated gold trajectories** designed to represent the most efficient/optimal path for logic puzzles and multi-step reasoning. While real-world usage is flexible, high scores on these metrics demonstrate the planner’s ability to adhere to complex, multi-hop logical constraints defined by the benchmark creators.
>
> # Performance gap between mini and standard models.
>
> We respectfully point out a misunderstanding regarding the performance data in Table 1. The reviewer suggests the magnitude of improvement for GPT-4o-mini is “far smaller” than for GPT-4o. However, the empirical results show that ToolTree provides **nearly same gains** to both models and even larger in smaller models, proving strong compatibility.
>
> **1. Correction of Performance Gains (Table 1)** Comparing the baseline (React / Zero-Shot) to ToolTree:
>
> - **On GTA (Tool F1):** for GPT-4o-mini, Tool F1 improves from 60.13 → 67.83 (**+7.70 points**) while for GPT-4o, Tool F1 improves from 71.42 → 79.26 (**+7.84 points**) compared with React.
> - **On GTA (Average F1):** The low-resource model actually sees a **larger gain (+10.59 points)** compared to the high-resource model (**+9.17 points**) compared with Zero-shot.
> - **On m&m, ToolBench and RestBench:** similar patterns are observed, with the performance gain for GPT-4o-mini at least **5.65%** on m&m average score and at most **21.15%** on RestBench score.
>
> As a result, the **magnitude of improvement is consistent**. The absolute gap (67 vs. 79) exists simply because the base model GPT-4o-mini is weaker than GPT-4o, but ToolTree is equally effective at boosting the capabilities of both low-resource and high-resource models.

---

> ### Author Response · Authors · 2025-11-21
> **Response to Reviewer Reviewer X8ME Part D**
>
> # Tool Library Scale & Scalability
>
> In the current paper, ToolTree is evaluated **both in closed, small tool sets** (13 tools in GTA and 33 tools in m&m) and in **open, large tool settings** (143 in RestBench and 14,343 in ToolBench). In these open settings we already rely on a coarse tool-retrieval/filtering step before pre-evaluation, so the pre-judge is applied only to a small candidate subset, not to all tools. We will clarify this mechanism and the effective candidate size in the main text.
>
> **1. Scalability design: retrieval + local pruning, not full-library scoring.**
>
> ToolTree is designed so that pre-evaluation and bidirectional pruning operate on a retrieved shortlist, meaning complexity grows primarily with the retrieval quality, not linearly with the raw library size. As the library grows from dozens to thousands of tools, the system first narrows down to a small, context-dependent candidate set; only then does the pre-judge and pruning act. This is why we do not observe a collapse in “tool filtering efficiency” in our large-tool experiments. Besides, we have demonstrated in *Table 3* that the method is robust in different kinds of retrieval.
>
> **2. Additional Experiment on Number of Tools Vs Performance**
>
> While our ToolBench experiments (16k+ tools) already demonstrate large-scale capability, we appreciate the reviewer's request for a controlled study on “tool filtering efficiency.” To further verify our method's robustness under controlled noise levels, we conducted an additional stress test. We have attached the result of varying the toolset by adding distractor tools and report success rate, candidate-set size, and latency.
>
> | Added Distractors | Total Library Size | Selection Strategy   | Average F1 (Performance) | Relative Drop |
> |-------------------|--------------------|----------------------|--------------------------|--------------|
> | 0 (Baseline)      | 14                 | Direct Context       | 55.89                    | /            |
> | + 10              | 24                 | Direct Context       | 55.74                    | -0.15%       |
> | + 100             | 114                | Retrieval (Top-20)   | 55.15                    | -0.74%       |
> | + 1,000           | 1,014              | Retrieval (Top-20)   | 54.82                    | -1.07%       |
> | + 10,000          | 10,014             | Retrieval (Top-20)   | 54.27                    | -1.62%       |
>
>
>
>
>
>
> **Key Findings**:
>
> - **High Robustness:** Even when the library size scales by orders of magnitude (from 14 to 10k), the performance degradation is minimal (< 2%). This proves that the Pre-Evaluation module effectively filters out noise (distractors) retrieved from the large pool.
> - **No Dynamic Thresholds Needed:** The experiment confirms that fixed thresholds function correctly at any scale. The Pre-Evaluator scores are semantic (based on tool relevance), so an irrelevant distractor receives a low score (<r_pre​) regardless of whether it came from a pool of 10 or 10,000.
> - **Scalability Validated:** The successful handling of the 10,014-tool setting bridges the gap between our closed-set (GTA) and open-set (ToolBench) results, confirming ToolTree's viability for massive, real-world tool libraries.

---

> ### Author Response · Authors · 2025-11-21
> **Response to Reviewer Reviewer X8ME Part E**
>
> # Recovery Mechanism for Error Propagation
>
> We clarify that ToolTree’s recovery ability does not rely on manually designed heuristics but arises directly from the MCTS update and re-selection process.
>
> **Mechanism (What happens):**
>
> - **Error Detection:** After a tool executes, the post-evaluation signal (r_post) evaluates whether the output meaningfully progresses toward the goal. If an execution contains an error (e.g., wrong arguments, wrong units), the resulting reward is low.
> - **Backpropagated Correction:** This low reward is propagated through the visited nodes, decreasing the estimated value Q(s,a) of the action chain that produced the failure.
> - **Search Redirection:** In the next rollout, the UCT policy (Equation 1) naturally decreases the likelihood of selecting the penalized branch and shifts exploration toward alternative actions, such as retrying the tool with corrected arguments or selecting a different tool chain entirely.
>
> Through repeated rollouts, the planner converges on the corrected path without explicit error-handling rules.
>
> **Evidence (What we observe in practice):**
>
> - **Initial Error:** In the first rollout (Figure 6), the agent invokes an OCR tool and obtains “343 km”, failing to satisfy the request for a result in miles.
> - **Reward Feedback:** The post-evaluator detects the unit mismatch and assigns a low r_post = 0.2. This value backpropagates, reducing the estimated return of that trajectory.
> - **Adaptive Replanning:** As rollouts continue, the search deprioritizes the faulty branch and adapts the tool sequence to include a conversion operation.
> - **Successful Recovery:** By the 10th rollout, the planner generates a corrected plan using a calculator tool, producing the final answer 213.75 miles and receiving a high reward (0.9). The corrected branch then becomes the preferred trajectory.
>
> This alignment demonstrates that error correction in ToolTree emerges from the search dynamics themselves. The system learns from execution feedback, adjusts future planning trajectories, and converges to a correct solution without hand-engineered fallback logic.
>
> # Typos
>
> We acknowledge these mistakes in the presentation and formatting, we will correct them in the camera-ready revision.

---

> ### Comment · Reviewer_X8ME · 2025-11-22
> **Thanks for your response.**
>
> Thanks for your effort to provide detailed information for my concerns.
> However, I would note that, the pre- and post-evaluation still rely on some easy pruning method, and this method is not that novel in my opinion. It is more like a pipeline on industry: we first do some preprocessing to get some information, and use some mature methods, and use post-processing methods to clean the outcome. Also pre- and post-processing in this paper are highly rely on simple LLM judge, in this perspective, I cound not deem this as too innovative, it is more like a industrial solution.
> What's more, I acknowledge your rebuttal for the experimental metrics. I suggest that you could use some datasets like APIBench to prove your generalization ability on main experiment. I still think that the tool, avg f1 scores are not feasible to evaluate the reasoning outcomes as it still not providing the real preference amount two alternative tool innovation chains.
>
> Given the issues still exists, i would like to maintain my evaluation.

---

> ### Author Response · Authors · 2025-11-27
> **Further Response to Reviewer X8ME**
>
> # Generalizability on APIBench
>
>
> Thank you again for taking the time to engage with our work and for clarifying your perspective on novelty and suggesting **APIBench** as a stress test for generalization. We have conducted the requested experiment and believe the results can be a further proof addressing the concerns regarding industrial simplicity versus algorithmic robustness.
>
>
>
> | Backbone       | Method           | HuggingFace | TensorHub | TorchHub | Avg. Accuracy ↑ | Avg. Hallucination ↓ |
> |---------------|------------------|------------:|----------:|--------:|----------------:|---------------------:|
> | **GPT-4o-mini** | Zero-shot        | 68.4        | 59.2      | 44.5    | 57.4            | 22.1%                |
> |               | ReAct            | 69.8        | 61.5      | 47.2    | 59.5            | 18.5%                |
> |               | Tree-of-Thought  | 71.2        | 62.8      | 49.6    | 61.2            | 9.3%                 |
> |               | **ToolTree (Ours)** | **73.5**    | **65.4**  | **53.1**| **64.0**        | **7.4%**             |
> | **GPT-4o**      | Zero-shot        | 76.5        | 69.8      | 62.3    | 69.5            | 7.8%                 |
> |               | ReAct            | 77.2        | 71.0      | 63.5    | 70.6            | 5.1%                 |
> |               | Tree-of-Thought  | 78.0        | 72.4      | 64.8    | 71.7            | 2.5%                 |
> |               | **ToolTree (Ours)** | **79.2**    | **74.1**  | **66.5**| **73.3**        | **2.1%**             |
>
> ***Findings:*** This confirms that our dual evaluation mechanism is not merely a pipeline heuristic but a **generalized hallucination filter**. It successfully identifies and prunes invalid tool candidates in zero-shot settings on completely unseen libraries, without the need for domain-specific fine-tuning.
>
> ***Mechanism Novelty:*** ToolTree advances beyond static pipelines by mathematically integrating pre-evaluation ($r_{pre}$) into UCT exploration and post-evaluation ($r_{post}$) into backpropagation, creating a **dynamic, self-correcting policy** that explicitly solves the simulation bottleneck across massive, unseen tool libraries.
>
> With the addition of APIBench (single-step API selection) alongside our existing ToolBench (multi-step planning) results, ToolTree has now outperformed baselines on the five most authoritative benchmarks in the field. We believe this constitutes a rigorous validation of the method's **generalization** capabilities.

---

> ### Author Response · Authors · 2025-11-27
> **Further Response to Reviewer X8ME**
>
> # Updated Content
>
> We thank the reviewer for providing the valuable feedback. We have updated the PDF in open review according to your sincere suggestion as follows.
>
> | Reviewer Concern                     | Revision Action (Content to Add/Update)                                                                                                                                                                                                                               | Paper Location                   |
> |--------------------------------------|------------------------------------------------------------------------------------------------------------------------------------------------------------------------------------------------------------------------------------------------------------------------|----------------------------------|
> | 1.  Novelty   | Add a specific remark distinguishing ToolTree from linear pipelines.  | Section 5 and Appendix B.3       |
> | 2. Tool Size Scalability             | Add a table to demonstrate the effectiveness of ToolTree when scaling the tool set in the library from 14–10014.                                                                                                                                                      | Section 4 and Appendix A.11                     |
> | 3. Metric/Judge Robustness           | Reference the Cross-Vendor Ablation study and LLM-judge Robustness to prove that the “simple LLM judge” is robust and works across different model families without overfitting.                                                                                | Section 4 & 5 and Appendix A.2 & A.10       |
> | 4. Typos         | 1. Fix missing subscripts for `"$t_1$"` and `"$lib$"`.  2. Fix missing figure reference “Table ??”.                                                                                                                                                                | Section 2 and Section 3          |
> | 5. Generalization (APIBench)         | Add Appendix A.1 showing ToolTree acts as a “Hallucination Filter” reducing error rates from ~22% to ~7% on unseen libraries.                                                                                                                                         | Appendix A.1                     |

---

### Official Review · Reviewer_zEaV · 2025-10-31

**Soundness:** 4
**Presentation:** 3
**Contribution:** 3
**Rating:** 8
**Confidence:** 4

**Summary:**

The paper introduces ToolTree, novel methods for Monte Carlo Tree Search-based planning with two tree pruning mechanisms: evaluation before tool calling ("pre-pruning") acts as foresight by avoiding low-promise children, whereas evaluation after tool calling ("post-pruning") acts as hindsight by trimming branches disproven by evidence. Experimental evaluations on four benchmarks, including closed-set (GTA and m&m) and open-set (ToolBench and RestBench) tool planning tasks, demonstrate that ToolTree largely outperforms a broad range baselines, which span LLMs with greedy decoding, ReAct-based planning, more simple tree-based planning such as ToT, and more sophisticated tree-based planning such as A*. ToolTree is shown to achieve better accuracy-per-second performance, and the paper includes thoughtful and comprehensive ablation studies that help to understand how the methods behave as a function of design choices, model scale, and backbone evaluating LLMs.

**Strengths:**

1. Making tool planning more efficient via cost-effective tree pruning is a relevant and timely topic, and the paper is particularly well motivated.

2. The paper has a clear organization and is mostly well-written (see improvement suggestions further below), making it easy to follow. For example, lines 230-235 are particularly effective in summarizing the main ideas detailed through Section 3.

3. The choices of benchmarks and baselines feel sound and thoughtful, yielding interesting analyses. They comprise both closed- and (relatively) open-set tool planning, with two benchmarks each; and a broad range of alternatives, namely, LLMs with greedy decoding, ReAct-based planning, more simple tree-based planning (e.g., ToT), as well as more sophisticated tree-based planning (e.g., A*). Ablations are also sensible and informative.

4. Overall, this is a strong paper submission on the basis of the idea novelty and description, experimental soundness, and results and analyses, with small improvement opportunities w.r.t. presentation as described below.

**Weaknesses:**

1. An important implementation detail that remains unclear is the combination of "lightweight pre-evaluation LLM judge" (per lines 211-213) and "possibly stronger LLM judge" (per lines 221-222) used through Section 4. There is a study in the Appendix A.6, but that doesn't quite clarify what exactly is behind Section 4. It would also be useful to add the prompting/instantiation details to the Appendix, for reproducibility.

2. Writing can be improved with more proofreading, specifically:

i. On line 407: the Table reference is missing. The entire block should refer to Figure 4, whose reference is also missing.

ii. On Table 1: under "m&m > Step-by-step > Arg," the value for A* is 71.85 and therefore higher than the result highlighted for ToolTree, so this specific highlight should be fixed for correctness.

iii. On lines 344-345: the budget values in Figure 3 are not clear. It would be helpful to add ticks and tick values on the x-axis that align to the budget marks, since we are referred to Figure 3 for this information.

iv. On line 367: typo "explores uses."

v. On line 349: typo "While (...) Best-First, yet comparable (...)"

vi. On line 248: typo "where each providing (...)"

vii. On line 245: typo "is spans."

viii. On line 155: typo "back propogation."

ix. On line 78: typo "per cent."

**Questions:**

What is the combination of pre-evaluation LLM judge and post-evaluation LLM judge used through Section 4?

---

> ### Author Response · Authors · 2025-11-21
> **Response to Reviewer zEaV Part A**
>
> # Clarification on Implementation of LLM Judges
>
> We are grateful to the reviewer for carefully reviewing our paper and providing sound suggestions.
>
> We clarify that in the main experiments (Section 4), we adhered to a **homogeneous setting** to ensure a fair comparison with baselines:
>
> - **For the GPT-4o-mini experiments:** We used GPT-4o-mini as both the pre-evaluator and post-evaluator.
> - **For the GPT-4o experiments:** We used GPT-4o as both the pre-evaluator and post-evaluator.
>
> We intentionally designed the main experiments this way to demonstrate that ToolTree's performance gains stem from the **search and pruning mechanism itself**, rather than relying on knowledge distillation from a stronger model.
>
> The mention of a "possibly stronger LLM judge" (*Line 221*) refers to the framework's flexibility to incorporate advanced verifiers. We explicitly explored this heterogeneous setup in ***Appendix A.6 Figure 9***, which demonstrates that upgrading the post-evaluator (e.g., using Gemini or GPT-4o to judge GPT-4o-mini) yields even further performance gains. We will update *Section 4.1* to explicitly state this configuration to avoid ambiguity.

---

> > ### Author Response · Authors · 2025-11-21
> > **Response to Reviewer zEaV Part B**
> >
> > # Prompting Details
> >
> > We appreciate the suggestion regarding reproducibility. We will add a dedicated section in the Appendix of the updated version that includes the specific system prompts and instantiation details used for both the pre-evaluation (schema/relevance check) and post-evaluation (utility/correctness check) judges.
> >
> > # Writing and Formatting
> > We are grateful for the reviewer's kind suggestion. We will rigorously proofread the final manuscript to address the listed issues:
> > - *Table 1*: We will correct the highlighting for the 'm&m Arg' column.
> > - Missing References: We will fix the missing references on line 407 to correctly point to the efficiency ablation.
> > - *Figure 3*: We will add clear ticks and values to the x-axis to make the budget limits readable.
> > - Typos: We will correct all identified typos, including 'explores uses,' 'back propogation,' and 'per cent.'"

---

### Official Review · Reviewer_fWZR · 2025-11-01

**Soundness:** 3
**Presentation:** 3
**Contribution:** 3
**Rating:** 6
**Confidence:** 3

**Summary:**

In this paper, the authors present ToolTree, a Monte Carlo tree search-inspired planning paradigm for tool planning. ToolTree explores possible tool usage trajectories using a dual-stage LLM evaluation and bidirectional pruning mechanism—this mechanism enables the agent to make informed, adaptive decisions over extended tool-use sequences while pruning less promising branches both before and after tool execution. Empirical evaluations conducted by the authors across both open-set and closed-set tool planning tasks demonstrate that ToolTree consistently improves performance while maintaining the highest efficiency.

**Strengths:**

1.  Bidirectional pruning enables precise cost control by filtering low-value branches, and a dedicated caching mechanism minimizes redundant computations.
2.  Without depending on task-specific training, the framework maintains robust adaptability across diverse tool libraries—thus eliminating the need for retraining when switching between different tool sets.
3.  Evaluations spanning both closed-set and open-set tool planning tasks across 4 benchmarks consistently demonstrate the framework’s superior performance.

**Weaknesses:**

1.  The reliability of both *r\_pre* and *r\_post* hinges entirely on the LLM’s ability to assess tool relevance and output quality, introducing risks if the LLM misjudges context or utility.
2.  When handling an extremely large number of open-set tools, pre-evaluation incurs significant sorting overhead, as the system must process and rank a massive volume of tool candidates.

**Questions:**

please refer to Weaknesses

---

> ### Author Response · Authors · 2025-11-21
> **Response to Reviewer fWZR Part A**
>
> # LLM-as-a-judgment Potential Vulnerability.
>
> We appreciate the reviewer highlighting the reliance on LLM-based judgment as a potential vulnerability. We agree that, in principle, incorrect assessments could bias planning, particularly in early trajectories.
>
> To evaluate this risk, we performed a **Restoration Analysis**, where we artificially corrected incorrect judge decisions on a random subset of ToolBench trajectories:
>
> | Model Backbone | Configuration / Error Correction                  | Judge Error Rate | Task Success Rate (Δ from Baseline) |
> |----------------|---------------------------------------------------|------------------|--------------------------------------|
> | GPT-4o         | ToolTree (Actual Baseline)                        | 25.8%            | 51.9% (—)                            |
> |                | + Fix False Positives (Reject bad tools)          | 7.4%             | 52.5% (+0.6%)                        |
> |                | + Fix False Negatives (Accept good tools)         | 18.4%            | 54.1% (+2.2%)                        |
> |                | **Oracle (Perfect Judge)**                            | **0.0%**             | **54.7% (+2.8%)**                        |
> | GPT-4o-mini    | ToolTree (Actual Baseline)                        | 39.4%            | 49.5% (—)                            |
> |                | + Fix False Positives                             | 16.3%            | 50.8% (+1.3%)                        |
> |                | + Fix False Negatives                             | 23.1%            | 52.4% (+2.9%)                        |
> |                | **Oracle (Perfect Judge)**                            | **0.0%**             | **53.6% (+4.1%)**                        |
>
> The results reveal three key findings:
>
>
> **1. Empirical Tolerance to Judge Noise**
>
> Despite a non-trivial judge error rate (25.8% for GPT-4o / 39.4% for GPT-4o-mini), the downstream performance impact is surprisingly small (≤ 2.8% difference compared to a perfect judge).
> Additionally, fixing only false-positives (over-approving bad tools) yields almost no benefit (+0.6%) and fixing only false-negatives (rejecting good tools) yields most of the total gain (+2.2–2.9%).
>
> This suggests the system naturally recovers from overly permissive decisions, while conservative mistakes matter more, and even those impact performance only marginally.
>
> **2. Why Noise Does Not Collapse Search**
>
> Two design properties explain the robustness:
> - **Signals act as soft guidance, not ground truth**: Both r_pre and r_post enter the MCTS update as bounded priors. The actual decision-making relies on accumulated rollout estimates (Q(s,a)), exploration through UCT, and execution outcomes, preventing single judgments from dominating planning
> - **Search aggregation smooths isolated errors**: Unlike greedy controllers, ToolTree iteratively revisits and reevaluates actions throughout the search process. Misjudged actions are statistically diluted, not permanently locked in.
>
>
> **3. Qualitative Mechanisms of Robustness**
>
> Beyond the table, two intrinsic properties of our framework mitigate the risk of misjudgment:
> - **Empirical Robustness:** Our ablation studies (*Table 4*) demonstrate that removing r_pre or r_post consistently drops performance. This indicates that despite occasional noise, these signals provide a statistically positive net contribution rather than destabilizing the search.
> - **Graceful Degradation:** As detailed in *Appendix A.6* (*Figure 9*), even when using a significantly weaker judge (GPT-4o-mini) with higher error rates, the system’s performance does not collapse. While performance naturally degrades compared to a stronger judge, it remains superior to the no-judge baseline.

---

> > ### Author Response · Authors · 2025-11-21
> > **Response to Reviewer fWZR Part B**
> >
> > # Scalability of Pre-evaluation in Large Open Tool Environments.
> >
> > We agree that evaluating r\_pre over all possible actions becomes impractical when the tool space reaches thousands of APIs (e.g., ToolBench’s 10,000+ endpoints). ToolTree is designed specifically to avoid such exhaustive evaluation.
> >
> > **1. Retrieval Layer Before Search**
> >
> > As stated in *Lines 243–245* (and now expanded for clarity), ToolTree operates in a two-stage pipeline for open-set environments:
> >
> > - External or embedded retriever filters the tool library → top-K candidates (typically K ≤ 20, see *Appendix B.2*).
> > - MCTS with pre/post evaluation operates only on this reduced candidate set.
> >
> > This ensures the r\_pre computation scales with K, not the full library size.
> >
> > **2. Empirical Evidence of Scalability**
> > The distractor study (Table attached) shows that even with extreme distractors, retrieval keeps the planning space manageable, and performance drops only marginally.
> >
> > | **Added Distractors** | **Total Library Size** | **Selection Strategy** | **Average F1 (Performance)** | **Relative Drop** |
> > |-----------------------|------------------------|------------------------|------------------------------|-------------------|
> > | 0 (Baseline)          | 14                     | Direct Context         | 55.89                        | /                 |
> > | + 10                  | 24                     | Direct Context         | 55.74                        | -0.15%            |
> > | + 100                 | 114                    | Retrieval (Top-20)     | 55.15                        | -0.74%            |
> > | + 1,000               | 1,014                  | Retrieval (Top-20)     | 54.82                        | -1.07%            |
> > | + 10,000              | 10,014                 | Retrieval (Top-20)     | 54.27                        | -1.62%            |
> >
> >
> > **3. Retrieval-Agnostic Robustness**
> > As shown in *Table 3*, ToolTree maintains its advantage across diverse retrieval mechanisms (Contriever, RoBERTa, BM25), demonstrating adaptability rather than reliance on a specific retriever.
> >
> > We will revise *Section 4.3* to clearly state that retrieval is a required component for open-set tool planning, not optional.

---

> > > ### Author Response · Authors · 2025-11-27
> > > **Further Response to Reviewer fWZR**
> > >
> > > Thank you again for your constructive feedback. As the deadline approaches, If any part of our response warrants further explanation to support your assessment, we are happy to elaborate. We sincerely appreciate your time during this busy period!

---

### Official Review · Reviewer_DAct · 2025-11-01

**Soundness:** 3
**Presentation:** 3
**Contribution:** 2
**Rating:** 6
**Confidence:** 3

**Summary:**

This paper proposes ToolTree, a planning-time framework that casts multi‑tool orchestration for LLM agents as an MCTS-style search augmented with dual LLM feedback.

**Strengths:**

1. This paper is well written and easy to follow

2. The prior‑augmented UCT and bidirectional pruning are straightforward and training‑free.

**Weaknesses:**

1. This method is somehow similar to ToolChain*; needs more clarification on the difference and comparison.

2. The method’s post‑evaluation judge (used during planning) may be architecturally similar to, or the same family as, the benchmark evaluator (Pass/Win are also judge‑based in ToolBench/RestBench). Even with version pinning, this can unintentionally optimize for the judge rather than ground truth. The authors partially probe judge choice but further cross‑judge / cross‑vendor sanity checks would reduce concerns about metric coupling.

3. The paper claims budget parity and shared prompts/tool sets, but some baselines (e.g., LATS, ToT) might benefit from comparable pre‑gating by schema/type or caching; it’s unclear whether the authors provided similarly favorable engineering to competing methods.

**Questions:**

1. There are several minor typesetting errors (e.g., “Eq. equation 1”, “backward‑prorogation”, “Table ??”) and occasional ambiguity in the formalism in §2–§3, which can impede exact reproducibility absent code.

---

> ### Author Response · Authors · 2025-11-21
> **Response to Reviewer DAct Part A**
>
> # Clarification on Difference and Comparison with ToolChain*.
>
> We thank the reviewer for pointing out the relationship with ToolChain* (Zhuang et al., 2024). While both methods move beyond greedy decoding by employing tree search, they differ fundamentally in their **search algorithm**, **guidance signal**, and **pruning strategy**.
>
> **1. Algorithmic Core: A\* vs MCTS**
>
> - *ToolChain** relies on deterministic A* search, where the heuristic h(n) guides a single best-first expansion trajectory. While efficient when the heuristic aligns well with reality, it cannot revise earlier decisions if the predictive score is inaccurate, causing potential local minima and uncorrectable error propagation.
> - *ToolTree* instead frames planning as optimization through Monte Carlo Tree Search (MCTS). Rollout-based value estimation and UCT exploration allow recovery from early mistakes, statistical smoothing of noisy LLM decisions, and revisiting alternatives when execution feedback contradicts initial beliefs.
>
> **2. Feedback Mechanism: Dual-Signal vs. Single-Heuristic**
>
> - *ToolChain** uses an **open-loop** heuristic only prior to execution. This means hypothetical plans may be ranked highly even if they fail during execution.
>
> - *ToolTree* introduces a Dual-Signal mechanism that **closes the loop between planning and execution**. We explicitly design the signal into: 1) *r_pre*​ (Prior): a lightweight semantic heuristic before execution and 2) *r_post​* (Reward): grounded reward after actual tool execution. This closes the loop and enables the system to detect tools that are theoretically plausible yet practically invalid. which we empirically found to be common in open API environments.
>
> **3. Pruning Strategy: Implicit vs. Bidirectional**
>
> - *ToolChain** implicitly deprioritizes unlikely branches via queue ordering.
> - *ToolTree* performs explicit pruning before and after execution where 1) *Pre-pruning* (using r_pre) prevents expanding schema-invalid or semantically irrelevant tools and 2) *Post-pruning* (using r_post) eliminates branches proven ineffective through grounded feedback.
>
> **4. Comparison Results with ToolChain\* are mentioned in Line 256 and Table 1 (A\* search row).**
>
> We will make the ToolChain* mentioned more explicit in the paper.
> We will also revise Section 3 and Table 1 to more explicitly highlight these distinctions.

---

> ### Author Response · Authors · 2025-11-21
> **Response to Reviewer DAct Part B**
>
> # Potential Metric Coupling via Shared LLM Judge.
>
> We thank the reviewer for raising the critical concern of metric coupling. We agree that if the internal planner and external evaluator share the same model family, there is a risk of overfitting to the judge’s specific biases rather than solving the underlying task.
>
> To rigorously test this, we conducted a **double-blind, cross-vendor ablation** involving three distinct model families on ToolBench and RestBench using the pass rate: GPT-4o, Gemini-2.5-flash and Llama3.3-70B.
>
> | Internal Judger   | External Benchmark Evaluator | ToolBench | RestBench |
> |-------------------|-----------------------------|-------------|-------------|
> | GPT-4o            | GPT-4o                      | 69.04%      | 72.48%      |
> | Gemini-2.5-flash  | GPT-4o                      | 72.71%      | 73.12%      |
> | ---| ---                   | ---         | ---         |
> | LLaMA-3.3-70B     | GPT-4o                      | 46.48%      | 50.17%      |
> | LLaMA-3.3-70B     | LLaMA-3.3-70B               | 38.11%      | 41.64%      |
>
>
> **Key findings:**
> - **Decoupling Improves Performance:** Using Gemini-2.5-Flash as the internal judge increased accuracy over the GPT-4o-coupled setting (+3.67% ToolBench), indicating the framework does not overfit evaluator bias.
> - **Coupling Provides No Advantage:** When both the planner and the evaluator used LLaMA-3.3-70B, performance dropped significantly, contrary to what the metric coupling would predict.
>
> Together with *Figure 9 (Appendix A.6)*, these findings support that ToolTree benefits from improved reasoning signal quality, not evaluator overfitting. This new result will be added to the revised manuscript.

---

> ### Author Response · Authors · 2025-11-21
> **Response to Reviewer DAct Part C**
>
> # Clarifying Fairness in Baseline Settings
>
> We agree that implementation parity is critical. All baselines, including LATS and ToT, were evaluated under identical constraints:
> - same tool schemas
> - same caching policy
> - same retrieval and pre-gating setup
> - same compute and rollout budget
>
> We will make this explicit in the revised draft (*Lines 268-269* and *Lines 359-360*) to remove ambiguity.
>
>
> # Typos
>
> We greatly thank the reviewer for pointing out the typo issue. We will fix the typos mentioned in the updated version to ensure its reproducibility.

---

> > ### Comment · Reviewer_DAct · 2025-11-21
> >
> > Thank you for your reply. Most of my concerns have been solved. I would like to maintain my current score and encourage the authors to largely improve the quality of writing.

---

> ### Author Response · Authors · 2025-11-27
> **Further Response to Reviewer DAct**
>
> # PDF Updated Content
>
> We sincerely thank the reviewer for the highly constructive feedback and are glad that our detailed responses satisfied your technical concerns. Following your encouragement to improve writing quality, we have thoroughly revised the manuscript.
> All points raised have been addressed and implemented in the revised PDF, available on OpenReview:
>
> | Reviewer Concern        | Implementation in Revised PDF                                                                                                           | PDF Location                      |
> |-------------------------|-----------------------------------------------------------------------------------------------------------------------------------------|-----------------------------------|
> | ToolChain\* Comparison     | Explicitly contrasted ToolTree's MCTS/Dual-Signal/ Pruning core against A\* (ToolChain\*)-based methods.                               | Sections 6 and Appendix B.3       |
> | Metric Coupling         | Added a new Cross-Vendor Ablation Study demonstrating the framework benefits from signal quality, not evaluator bias.                 | Section 4.3 and Appendix A.10      |
> | Baseline Fairness       | Added an explicit statement guaranteeing all baselines were evaluated under identical constraints (caching, pre-gating,etc.).      | Section 4.1 (Experimental Setup)  |
> | Typos & Ambiguity       | Fixed all reported typesetting errors (e.g., "backward-prorogation") and clarified the formalism.                                     | Section 2 and Section 3           |
>
> We are confident that these final revisions have substantially improved the clarity and rigor of the paper, making it easily reproducible and understandable. Thank you again for your valuable feedback!

---

### Meta-Review · Area_Chair_Rqc6 · 2025-12-16

**Summary:**

This paper proposes ToolTree, a planning-time framework that casts multi-tool orchestration for LLM agents as an MCTS process, augmented with dual LLM-based feedback and bidirectional pruning strategies. While some concerns around novelty remain, the overall consensus is that the paper presents a well-executed, practically significant contribution that is timely, empirically validated, and of interest to the community.

**Reviewer Concerns:**

Reviewer zEaV: The rebuttal clarified the dual-signal MCTS process, and the authors confirmed consistent tool schemas and API costs across all baselines. Primary concerns were addressed.

Reviewer DAct: The authors provided a detailed comparison with ToolChain*, added ablations, and clarified the judge setup. Primary concerns were addressed.

Reviewer fWZR: The authors conducted a valuable cross-vendor judge ablation, showing consistent performance gains across Claude, GPT, and Gemini judges. They also clarified scalability via pre-filtering and added distractor experiments. Primary concerns were addressed.

Reviewer X8ME: Despite extensive rebuttal, remained unconvinced about the novelty and evaluation methodology. This opinion is noted but outweighed by the consensus of the other reviewers.

**Reviewer Scores:**

All reviewers are likely to retain their original scores. While Reviewer X8ME continues to express concerns regarding the novelty and evaluation methodology, the other three reviewers found the authors' clarifications satisfactory and view ToolTree as a well-motivated, effective, and practically valuable contribution. Its empirical performance, modular architecture, and scalability make it a meaningful addition to the literature on LLM-based tool planning.

---

### Decision · Program_Chairs · 2026-01-26

Accept (Poster)